# Continuous addition of progenitors forms the cardiac ventricle in zebrafish

Anastasia Felker[1], Karin D. Prummel[1], Anne M. Merks [2], Michaela Mickoleit[3], Eline C. Brombacher[1], Jan Huisken [3,4], Daniela Panáková [2,5] & Christian Mosimann [1]

The vertebrate heart develops from several progenitor lineages. After early-differentiating first heart field (FHF) progenitors form the linear heart tube, late-differentiating second heart field (SHF) progenitors extend the atrium and ventricle, and form inflow and outflow tracts (IFT/OFT). However, the position and migration of late-differentiating progenitors during heart formation remains unclear. Here, we track zebrafish heart development using transgenics based on the cardiopharyngeal gene *tbx1*. Live imaging uncovers a *tbx1* reporter-expressing cell sheath that continuously disseminates from the lateral plate mesoderm towards the forming heart tube. High-speed imaging and optogenetic lineage tracing corroborates that the zebrafish ventricle forms through continuous addition from the undifferentiated progenitor sheath followed by late-phase accrual of the bulbus arteriosus (BA). FGF inhibition during sheath migration reduces ventricle size and abolishes BA formation, refining the window of FGF action during OFT formation. Our findings consolidate previous end-point analyses and establish zebrafish ventricle formation as a continuous process.

[1] Institute of Molecular Life Sciences, University of Zürich, Winterthurerstrasse 190, 8057 Zürich, Switzerland. [2] Max Delbrück Center for Molecular Medicine in the Helmholtz Association, Robert-Rössle-Str. 10, 13092 Berlin, Germany. [3] Max Planck Institute of Molecular Cell Biology and Genetics, Pfotenhauerstrasse 108, 01307 Dresden, Germany. [4] Morgridge Institute for Research, 330N Orchard St, Madison, WI 53715, USA. [5] DZHK (German Center for Cardiovascular Disease), Partner Site Berlin, Berlin 10115, Germany. Correspondence and requests for materials should be addressed to D.Páká. (email: daniela.panakova@mdc-berlin.de) or to C.M. (email: christian.mosimann@imls.uzh.ch)

Vertebrate cardiomyocytes derive from an early vs. a late-differentiating progenitor pool within the anterior lateral plate mesoderm (ALPM) that can be divided into first heart field (FHF) and second heart field (SHF)[1]. After the early-differentiating FHF assembles the linear heart tube that in mammals forms the left ventricle and parts of the atria, the late-differentiating SHF contributes to the atria, right ventricle, and OFT[2–5]. In mice, the SHF forms within the medial and posterior epithelial-like field in the splanchnic ALPM on either side of the FHF-derived cardiac crescent and is detectable through expression of markers including Fgf10 and Isl1[5–7]. Defects in SHF contribution to the heart lead to a broad variety of congenital heart malformations affecting the arterial pole[8]. The original purpose of a second, late-forming wave of the myocardium remains unknown. An emerging concept places the SHF in close developmental and evolutionary lineage relationship with pharyngeal and head muscle progenitors: in this context, the cardio-pharyngeal field (CPF) is defined across chordate species as a domain within the splanchnic or pharyngeal ALPM harboring SHF and branchiomeric progenitor cells[9]. Consequently, a key aspect of the SHF is its cardiac specification coordinated with head muscle differentiation[10].

The basic mechanisms of vertebrate heart formation are evolutionarily conserved. Typical for teleosts, the zebrafish heart consists of two chambers, an atrium and a ventricle (Fig. 1a). At the arterial pole, the myocardium transitions into laminin-rich myocardium referred to as conus arteriosus (CA) followed by the elastic bulbus arteriosus (BA) that functions as a smooth-muscle-based pressure capacitator similar to the mammalian aortic arch[11,12]; the CA together with the BA is commonly, but not consistently, defined as OFT[12,13] (Fig. 1a). Cardiac progenitors become detectable in the zebrafish ALPM by bilateral expression of several conserved cardiac transcription factors, including nkx2.5, hand2, and gata4/5[14–17]. By 14–18 h post fertilization (hpf), these bilateral progenitors condense at the midline as the cardiac disk that forms the cardiac cone; the subsequently emerging linear heart tube consists of the endocardium and the surrounding early-differentiating cardiomyocytes referred to as FHF myocardium, discernible at 16–18 hpf by differentiation markers including myl7[18–20]. Myocardial expression of drl-based transgenes selectively marks FHF-derived cardiomyocytes from late somitogenesis[21]. Starting from 26 hpf, akin to the mammalian heart, a late-differentiating wave of prospective cardiomyocytes and smooth-muscle cells extends the venous IFT and arterial OFT of the beating zebrafish heart, referred to as SHF lineage[22–24] (Fig. 1a). The zebrafish heart therefore recapitulates key processes of multi-chambered heart formation[20].

Lineage tracing, several molecular markers including ltbp3, isl1, and mef2c, or the maturation speed of fluorescent reporters have been used to describe late-differentiating SHF-like myocardium in zebrafish[22–27]. Nonetheless, the lineage separation and developmental connection between the ventricular myocardium and OFT structures remain vaguely defined. Further, the position of SHF-assigned cells during formation of the linear heart tube has remained uncertain. Genetic lineage tracing has shown that both the distal ventricle and OFT derive from nkx2.5- and ltbp3-reporter-expressing cells[24,28]. nkx2.5:Kaede-based lineage tracking has indicated that most of the ventricular myocardium is already condensed at the cardiac disk, but whether all these cells then migrate with the emerging linear heart tube or stay behind has remained unresolved[29]. nkx2.5:Kaede further marks a seemingly distinct group of cells posterior and outside of the forming heart tube that also contributes myocardial progenitors to the distal ventricle and OFT[30]; how these cells connect to the other ventricular progenitors remains to be uncovered. Furthermore, position-based cell-labeling experiments have mapped BA origins

to the medio-central region of the heart-forming ALPM that corresponds to the expression domain of nkx2.5 and gata4[23], with the proximal-most part of the BA arising from nkx2.5 reporter-expressing pharyngeal arch 2 mesoderm[29]. Altogether, these analyses support a model of addition of the majority of late-differentiating myocardium to the ventricle and BA formation after establishment of the linear heart tube.

The T-box transcription factor Tbx1 is expressed within the CPF of various chordates and directs cardiac development by maintaining proliferation and suppressing differentiation of SHF cardiac progenitor cells[9,31–33]. Impaired TBX1 function in humans results in DiGeorge syndrome[32] with variable cardiac defects, including tetralogy of Fallot, OFT defects, and an interrupted aortic arch; defects that are recapitulated in Tbx1-mutant mice[34–36]. The zebrafish tbx1 mutant van gogh (vgo) displays among other phenotypes also defects in the pharyngeal arches and a smaller BA, underlining the conserved function of Tbx1 in CPF control[37,38]. The 12.8 kilobases (kb) upstream of the murine Tbx1 gene as a transgenic reporter principally recapitulate endogenous expression through separable Forkhead factor-recruiting enhancers that drive pharyngeal/anterior endoderm vs. mesoderm expression, including activity in the OFT[39–41]. While these enhancers are sufficient in transgenic reporters, endogenous Tbx1 expression is redundantly coordinated by additional elements in the vicinity of the locus[42].

Here, we isolate cis-regulatory elements from the zebrafish tbx1 locus to visualize the dynamics of ventricle and OFT formation. Combining selective plane illumination microscopy (SPIM) imaging with genetic and optogenetic lineage tracing, we capture the formation of the linear heart tube with concomitant migration of an undifferentiated sheath of tbx1 reporter-expressing cells that are continuously added to, and gradually differentiate at, the arterial pole of the heart. Meanwhile, BA progenitors reside in the tbx1 reporter-positive pharyngeal ALPM and migrate later toward the late-differentiating distal pole of the ventricle to become smooth muscle cells. Combining chemical and genetic perturbations, we find a distinct temporal requirement for FGF signaling in controlling ventricle and BA size vs. BA specification. In contrast to models that postulate a distant cellular origin and stepwise addition of SHF cells, our findings establish that incorporation of the zebrafish ventricle and OFT structures into the linear heart tube is a continuous process with distinct phases of FGF activity during CPF differentiation.

## Results

**Activity of zebrafish tbx1 regulatory elements.** In our ongoing efforts to isolate cis-regulatory elements active within the lateral plate mesoderm (LPM) to observe cardiovascular cell fate partitioning, we generated zebrafish tbx1 reporter transgenics based on the high ranking of tbx1 expression in transcriptome analysis of zebrafish LPM (within top-20 enriched genes)[21]. Transgenic reporters in mice have established core regulatory elements sufficient for recapitulating Tbx1 expression[41]. Consistently, we observed specific EGFP reporter activity driven by the 3.2-kb upstream region of zebrafish tbx1 in embryos carrying transgenic insertions of Tg(–3.2tbx1:EGFP), with minimal variability between six individual transgenic lines (Fig. 1b, Supplementary Fig. 1, Supplementary Table 1).

In late epiboly, tbx1:EGFP akin to endogenous tbx1 expression broadly labels a dorsal/anterior domain (Fig. 1c, Supplementary Fig. 1). During somitogenesis, tbx1:EGFP expression is detectable in anterior bilateral domains (Fig. 1d, e) and at 36 hpf in the pharyngeal arches and in the heart (Fig. 1f). While we do not detect significant endogenous tbx1 mRNA expression in the heart consistent with previous reports[43], we readily observed tbx1:

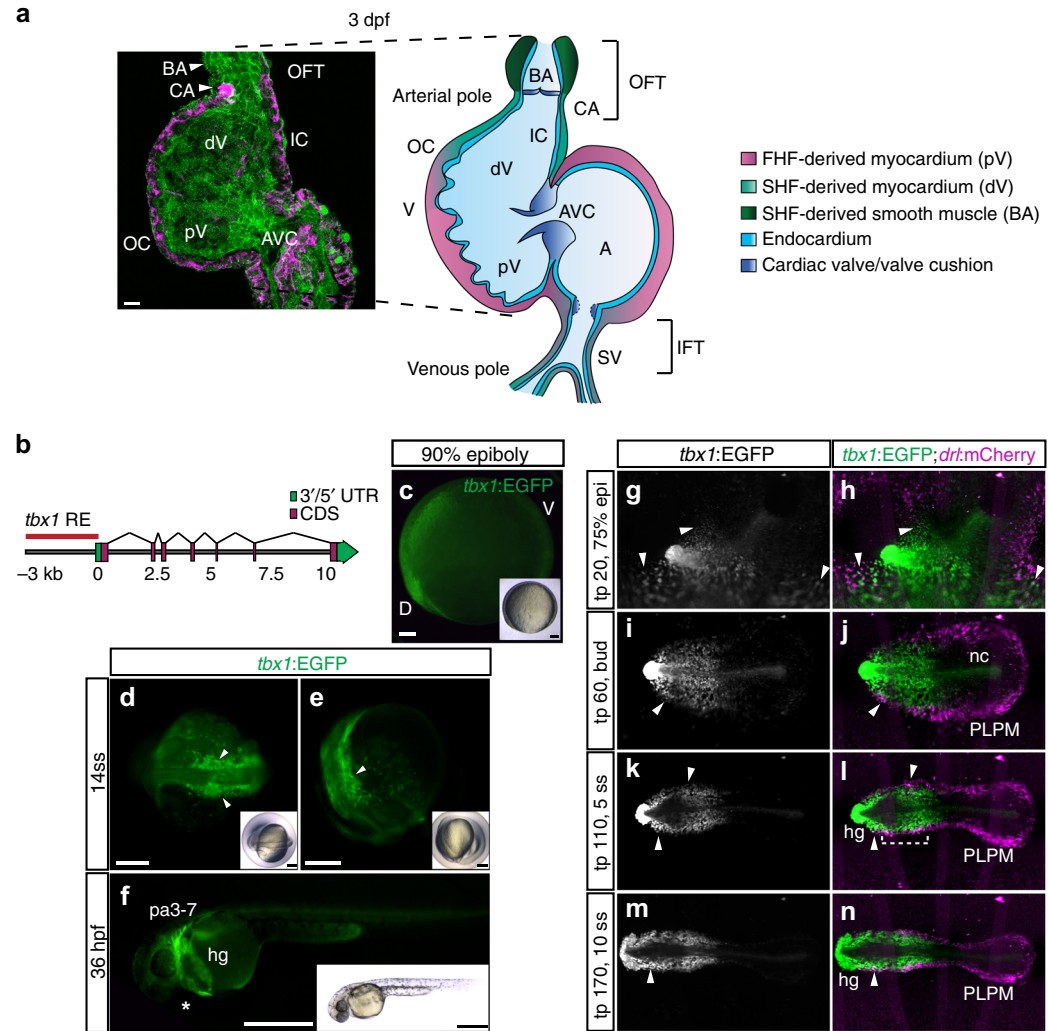

**Fig. 1** *tbx1* reporter expression and lineage contribution in the cardiopharyngeal field and ALPM. **a** Confocal *Z*-projection and schematic representation of a 72-hpf zebrafish heart with two chambers, the ventricle (V) and atrium (A) separated by a valve at the atrioventricular canal (AVC). The isolated heart is stained for MHC marking the myocardium (magenta) and a-PKC marking all cells (green). The FHF-assigned myocardium contains the proximal ventricle (pV) and the majority of the atrium (A), SHF-assigned myocardium forms the distal ventricle (dV) and outflow tract (OFT), shown in magenta and green in the schematic, respectively. The OFT includes the conus arteriosus (CA), comprising the myocardial connection of the ventricle to the bulbus arteriosus (BA), and the smooth muscular BA itself. The lineage contributions to the sinus venosus (SV)/inflow tract (IFT) and developmental timing of IFT valve formation remain unresolved. IC: inner curvature, OC: outer curvature. **b** Genomic locus of the zebrafish *tbx1* gene; the red line indicates a 3.2-kb *cis*-regulatory region amplified to drive reporter transgenics. **c** Representative *tbx1*:EGFP reporter transgene expression in a dorsal/anterior field during gastrulation (90% epiboly, dorsal/anterior to the left). **d, e** *tbx1*:EGFP at 14 ss; dorsal (**d**) and lateral views (**e**) of the prospective cardiopharyngeal field (CPF, arrowheads). **f** Lateral view of a 36-hpf *tbx1:EGFP* embryo with EGFP expression in the CPF-derived pharyngeal arches (pa3–7) and heart (asterisk). EGFP expression also marks the prechordal mesoderm-derived hatching gland (hg). **c–f** Insets depict bright-field images of the respective fluorescent images. **g–n** Mercator projection of representative stages from panoramic SPIM-imaged *tbx1:EGFP;drl:mCherry* double-positive transgenic embryos (*n* = 3); dorsal views. *tbx1*:EGFP expression is confined to the anterior of the embryo, with no EGFP signal in the posterior LPM (PLPM). Expression in the notochord (nc) and hatching gland (hg) is likely related to early prechordal plate activity of the reporter. Note the double-positive cells at the outermost domain of the *tbx1*:EGFP-positive anterior cell population (arrowheads and bracket). Scale bars 10 μm (**a**), 100 μm (**c**), 200 μm (**d, e**), and 500 μm (**f**). **g–n** Anterior to the left

EGFP expression in cardiac precursors, indicating distinct dynamics of our reporter compared to endogenous *tbx1* akin to mouse *Tbx1* reporters (Supplementary Fig. 1)[39–41]. To resolve *tbx1* reporter expression in relation to the *drl*-labeled LPM[21], we performed in toto panoramic SPIM imaging[44] on *tbx1*:EGFP;*drl*:mCherry transgenic embryos from gastrulation (70% epiboly) to mid-somitogenesis (15 hpf) (Supplementary Movie 1). At late epiboly to tailbud stages, when the LPM condenses around the embryo margin, we detected overlapping *tbx1*:EGFP expression in *drl*:mCherry-expressing LPM cells, medial within the ALPM

and lateral-most within the *tbx1* reporter-expressing domain (Fig. 1g–j). These cells condensed further at the margin of the *tbx1*:EGFP-expressing domain throughout early somitogenesis (Fig. 1k–n). Thus, a subpopulation of the *tbx1* reporter-expressing cells represents ALPM, while the medial field of *tbx1* reporter-positive cells potentially represents endoderm precursors, as we observe the labeling of endodermal derivatives (i) in 3 days post fertilization (dpf) *tbx1*:EGFP-expressing embryos, as well as (ii) with CreERT2/*lox*-mediated lineage tracing with *tbx1:creERT2* (*Tg(−3.2tbx1:creERT2)*[zh703]) and *hsp70l:Switch* (*Tg(−1.5hsp70l:*

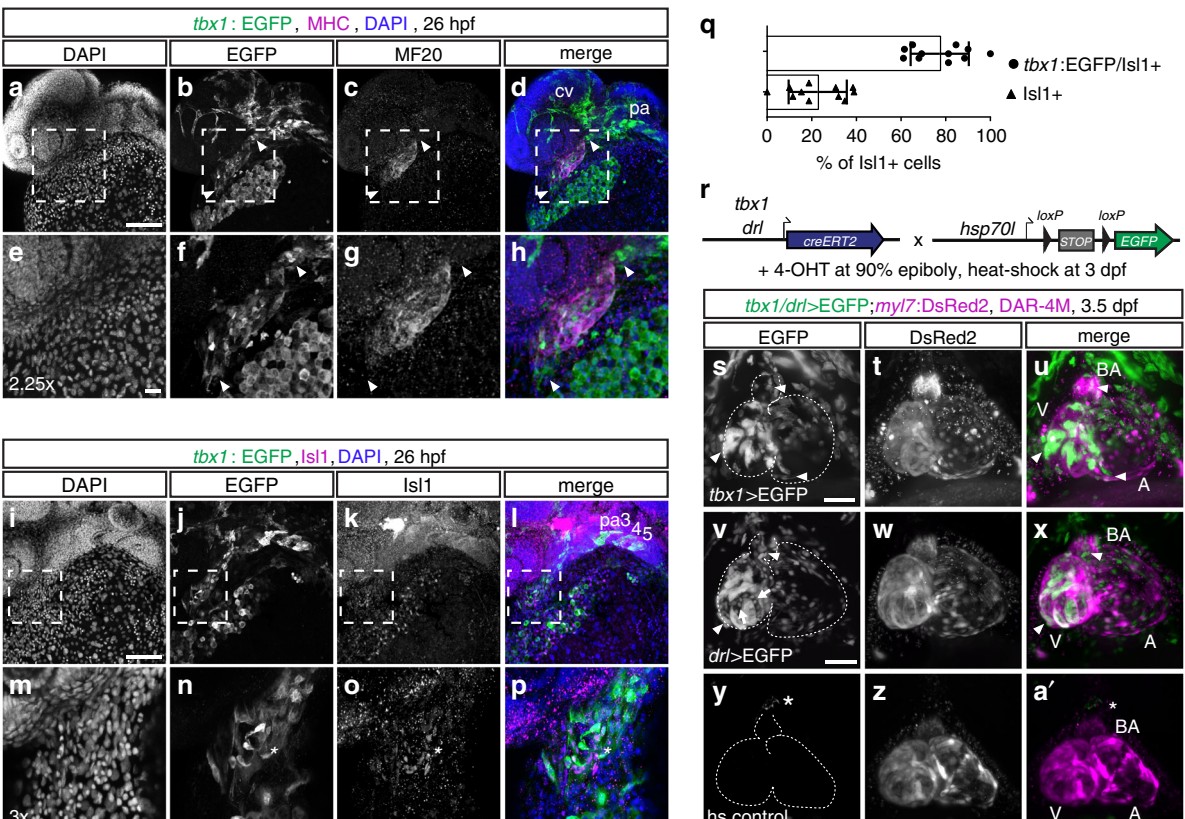

**Fig. 2** *tbx1*+ cells contribute to LPM-derived cardiac lineages. **a–p** Representative maximum intensity projections of whole-mount *tbx1*:EGFP-expressing embryos counterstained for anti-EGFP and anti-MHC (*n* = 3) (**a–h**) or anti-Isl1 (*n* = 11) (**i–p**) at 26 hpf; lateral views, anterior to the left. **a–d** *tbx1* reporter expression can be detected in the MHC-positive linear heart tube and in the MHC-negative poles at the cardiac inflow and outflow tracts (arrowheads); **e–h** depicts a 2.25x magnification of the framed area in **a–d**. *tbx1*:EGFP also marks the pharyngeal arches (pa) and endothelial cells of the cranial vasculature (cv) (**d**). **i–p** *tbx1* reporter-expressing cells at the IFT co-express the SHF marker Isl1 (asterisks **n**, **o**); **m–p** depicts a 3x magnification of the framed area in **i–l**. **q** Quantification of *tbx1*:EGFP/Isl1 double- compared to Isl1 single-positive cells at the IFT of the linear heart tube, *n* = 11 individual embryos analyzed, means ± s.d. **r** Lineage tracing of *tbx1* and *drl* reporter-expressing cells, shown in representative embryos. *tbx1:creERT2* (*n* = 11) or *drl:creERT2* (*n* = 3) transgenics, respectively, were crossed to the ubiquitous *hsp70l:Switch loxP* tracer line, embryos were 4-OHT-induced at 90% epiboly, and heat-shocked at 3 dpf. **s–a′** Live SPIM imaging of still hearts of representative lineage-traced and control embryos; maximum intensity projections of ventral views, anterior to the top, dashed outlines mark the heart with the bulbus arteriosus (BA), atrium (A), and ventricle (V). **s–u** *tbx1:creERT2* lineage tracing (*tbx1* > EGFP) at late gastrulation labels *myl7*:DsRed2-expressing cardiomyocytes (arrowheads) in the ventricle and inflow tract of the atrium, and DAR-4M-stained cells (arrowhead) in the BA. **v–x** *drl:creERT2*-mediated lineage tracing (*drl* > EGFP) at 90% epiboly marks all cardiomyocytes (arrowheads) in the ventricle and atrium, BA cells (arrowhead), and the endocardium (arrows). **y–a′** *tbx1:creERT2;hsp70l:Switch* transgenics without 4-OHT treatment and heat-shocked at 3 dpf show no specific EGFP expression (asterisks mark the autofluorescent pigment cell). Scale bars 20 µm (**e–h**, **m–p**), 100 µm (**a–d**, **i–l**, **s–a′**)

*loxP-STOP-loxP-EGFP,cryaa:Venus)^(zh701)* transgenics (Supplementary Fig. 2). Moreover, cranial cartilage is labeled by *tbx1* reporter expression and with *tbx1:creERT2*-mediated lineage tracing, suggesting additional *tbx1* reporter expression in neural crest lineages (Supplementary Fig. 2). Taken together, transgenic zebrafish reporter expression based on the upstream 3.2-kb *tbx1* *cis*-regulatory region approximates key aspects of reported Tbx1 activity[9,31,45] and visualizes a dynamic anterior endoderm and mesoderm domain that includes ALPM.

**tbx1 reporter cells contribute to venous and arterial poles**. To resolve cardiac *tbx1*:EGFP expression, we analyzed *tbx1:EGFP* transgenics co-stained for the differentiated cardiomyocyte-expressed myosin heavy chain 1E (MHC) (Fig. 2a–h). At 26 hpf, when the differentiating cardiomyocytes in the linear heart tube represent FHF derivatives[21,27], we detected *tbx1* reporter expression in most of the differentiated ventricular cardiomyocytes and additionally in two MHC-negative domains at the IFT and OFT (Fig. 2a–h). At 26 hpf, we detected that at the IFT of the

linear heart tube, on average, 77.3% of *tbx1*:EGFP cells are positive for Isl1, while only 22.7% are Isl1-positive alone (*n* = 11; Fig. 2i–p, q, Supplementary Movie 2), consistent with inferred SHF identity of IFT cells[25,27,46].

To corroborate which cardiac lineages form from *tbx1* reporter-expressing cells, we performed genetic lineage tracing with *tbx1:creERT2* (Fig. 2r). 4-OHT induction of CreERT2 starting from shield stage to 90% epiboly (6–9 hpf) labeled ventricular cardiomyocytes, including the distal ventricle and OFT region, scattered atrial cells around the IFT, and the diaminorhodamine-4M AM (DAR-4M)-reactive smooth-muscle cells in the BA[13] (Fig. 2s–u, y-a′, Supplementary Fig. 3). These cardiac descendants of *tbx1* reporter-expressing cells likely derive from ALPM: pan-LPM lineage tracing using *drl:creERT2* from 90% epiboly broadly marks atrial and ventricular myocardium plus the DAR-4M-stained BA (Fig. 2v–x, y-a′, Supplementary Fig. 3), in line with previous LPM labeling by *drl:creERT2*[21,47] and selective ALPM tracing of OFT structures[28,29]. While *drl*-mediated ALPM lineage tracing labeled the endocardium, cranial vessels (Fig. 2s–u,

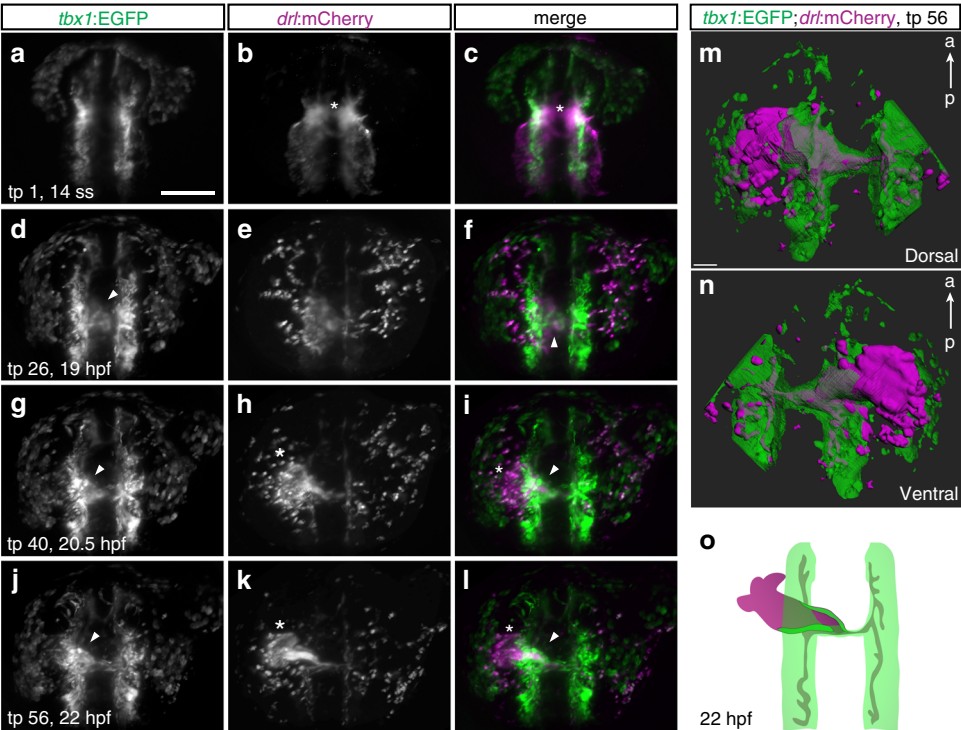

**Fig. 3** A *tbx1+* sheath forms at the base of the FHF-derived heart tube. **a–l** Maximum intensity projections of representative stages from SPIM-imaged *tbx1: EGFP;drl:mCherry* double-positive transgenic embryos; dorsal views, anterior to the top. Imaging was initiated at 14 ss (16 hpf) and cardiac development followed until linear heart tube (LHT) stage (22–23 hpf, n = 3). **a–c** 14 ss stage embryo at the onset of medial FHF migration (asterisk). **d–f** The forming cardiac disk already contains *tbx1*:EGFP-positive cells (arrowhead). **g–l** *tbx1*:EGFP-positive cells (arrowhead in **g**, **i**) assemble at the base of the extending *drl* reporter-expressing heart cone (asterisk in **h**, **k**) and are contained in the LHT (arrowhead in **j**, **l**), note the absence of *tbx1*:EGFP reporter-expressing cells at the leading edge (asterisk in **k**, **l**) of the forming heart tube. **m**, **n** 3D segmentation (dorsal and ventral view) revealing a *tbx1* reporter-expressing sheath of cells engulfing the *drl* reporter-expressing endocardium at 22–23 hpf. **o** Schematic of *tbx1* and *drl* reporter-expressing cell arrangements at the end of imaging. Scale bars 50 μm (**m**), 200 μm (**a–l**)

Supplementary Fig. 3), and several head muscle groups (Supplementary Fig. 3), *tbx1* reporter expression and lineage tracing labeled the same structures without the endocardium, but additionally also craniofacial cartilage and endoderm derivatives, in line with Tbx1 as a CPF marker[9] (Fig. 2d, s, Supplementary Figs. 2, 3). These results indicate that the *tbx1* reporter expression domain entails ALPM-derived cells that contribute to cardiac lineages including IFT and OFT structures.

**tbx1+ progenitors migrate with the linear heart tube.** By end-point analysis in zebrafish, different positions and migration dynamics have been assigned to IFT-, ventricle-, and OFT-contributing undifferentiated SHF progenitors, including delayed migration from pharyngeal ALPM[22,23], condensation with the forming cardiac disk[29], and localization posterior to the forming cardiac cone[30]. To visualize the cardiac *tbx1* reporter-expressing cells, we SPIM-imaged *tbx1:EGFP;drl:mCherry* embryos from 16 hpf (14 ss) to 22–24 hpf to capture all stages of heart field migration up to the onset of heartbeat (Supplementary Movie 3). Of note, at these stages, *drl*:mCherry expression in the heart field gradually confines from previous pan-LPM expression to restricted expression in FHF-derived lineages and robust expression in the endocardium[21]. At 14 ss, the *drl*:mCherry-expressing heart field is arranged as a bilateral LPM domain as is the nearly overlapping *tbx1*:EGFP-expressing field before condensing at the midline (Fig. 3a–f). From approximately 16 ss onward, while midline-centered migration and formation of the *drl*:mCherry-positive cardiac disk completed, *tbx1*:EGFP-positive cells contributed to the cardiac disk (Fig. 3d–f) and further disseminated from the bilateral *tbx1* reporter domain to migrate

medially along *drl*-positive endothelial progenitors (Fig. 3g–i). At 22 hpf, *tbx1* reporter-positive cells formed a sheath of cells at the prospective arterial pole of both *tbx1* and *drl* reporter-expressing linear heart tube (Fig. 3j–l). 3D segmentation confirmed that in its entirety, the *tbx1* reporter-positive cell population surrounded the growing endocardium like a sleeve trailing outward to the still bilateral progenitors (Fig. 3m–o, Supplementary Movie 4). These observations suggest co-migration of *tbx1* reporter-expressing cells as part of the forming, jogging linear heart tube and continuous with the prospective arterial pole (Fig. 3m–o).

**The tbx1+ domain is continuous with the early myocardium.** To understand the formation of the arterial pole of the heart tube in greater detail, we imaged *tbx1:EGFP;myl7:DsRed2* double-transgenic embryos from 18 to 30 hpf (Supplementary Movie 5, Supplementary Fig. 4). The slow folding of DsRed2 during this time frame discriminates between the early differentiated, DsRed2-positive (FHF-assigned), and the later-differentiating, DsRed2-negative (SHF-assigned) *myl7*-expressing cardiomyocytes[26,27]. Correspondingly, we detected *myl7*:DsRed2 expression in the migrating and differentiating heart tube from 24 hpf onward (Supplementary Fig. 4). We observed *tbx1* reporter-expressing cells at the arterial pole of the heart tube (i) that based on the absence of *myl7*:DsRed2 expression throughout 24–30 hpf correspond to the later-differentiating SHF myocardium, and (ii) that were connected to, and migrated together with, the *myl7*-positive myocardium (Supplementary Fig. 4).

To overcome the complications of live-tracking associated with the onset of heartbeat at 24 hpf with continued ventral and rostral heart tube migration[21–23,28], we performed SPIM-based high-

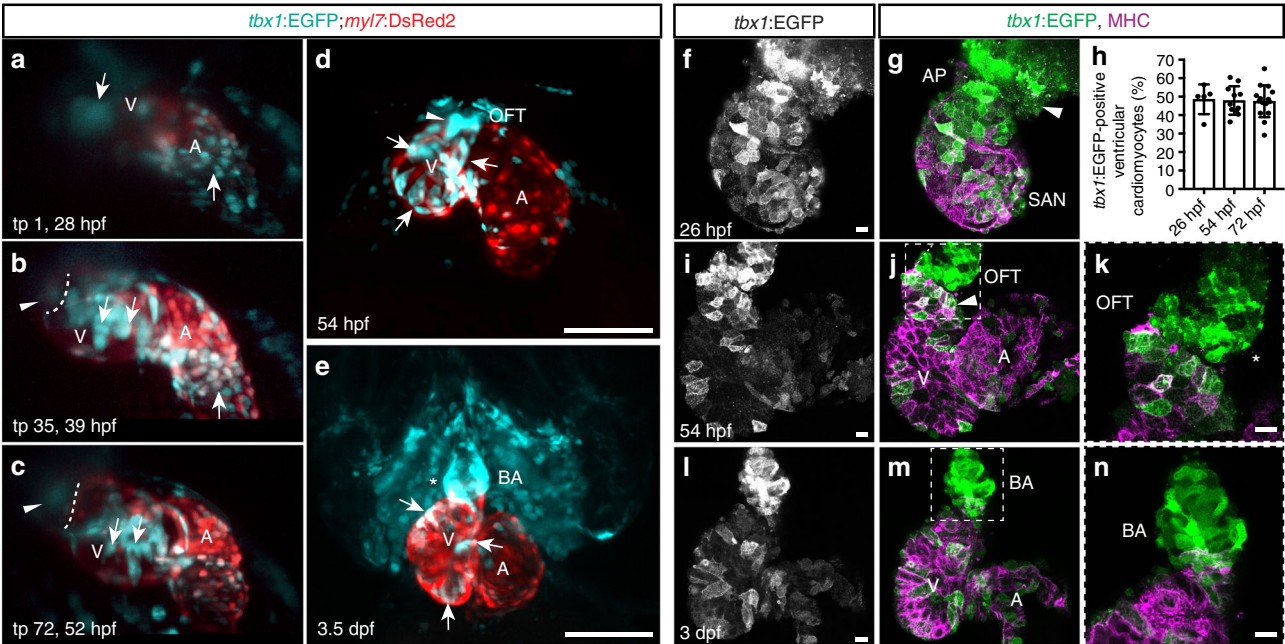

**Fig. 4** *tbx1+* myocardial precursors connect to the FHF myocardium during heart tube stages. **a–c** Maximum intensity projections of representative stages of a high-resolution reconstruction of the beating heart of a *tbx1*:EGFP;*myl7*:DsRed2 double-positive transgenic between 28 and 52 hpf; lateral view (right side) of the embryo, anterior to the top, ventricle to the upper left, atrium to the lower right, and cardiac imaging phase 27 ($n = 1$); ventricle (V), atrium (A), and bulbus arteriosus (BA). Arrows indicate *tbx1+/myl7-* cells at the OFT and IFT at the beginning of the time-lapse (**a**) that gradually turn on *myl7* reporter expression (**b**, **c**). The dashed line (**b**, **c**) indicates the distal end of the ventricle and the arrowheads point to *tbx1*:EGFP-expressing cells at the OFT that never cross into the ventricle and are likely BA precursors. **d, e** Maximum intensity projections of SPIM-imaged *tbx1*:EGFP;*myl7*:DsRed2 double-positive transgenic hearts stopped from contracting with BDM at 54 hpf ($n = 3$) or 3.5 dpf ($n = 7$), respectively; ventral views, anterior to the top. *tbx1* reporter expression can be detected in differentiated *tbx1/myl7* reporter double-positive cardiomyocytes (arrows **d**, **e**), the *tbx1+/myl7-* OFT at 54 hpf (arrowhead **d**), and the *tbx1+/myl7-* BA at 3.5 dpf (asterisk **e**). **f, g, i–n** Top-down 2-μm confocal section of isolated zebrafish hearts at 26 (**f**, **g**), 54 (**i–k**), and 72 hpf (**l–n**) from *tbx1*:EGFP, counterstained with anti-GFP and anti-MHC; OFT/BA to the top, sinoatrial node (SAN) or atrium (A) to the bottom left, and ventricle (V) to the top left. **k, n** A magnification of the framed area in **j**, **m**. **f, g** *tbx1*:EGFP is expressed at the MHC-negative arterial pole (AP) of the heart tube (arrowhead). **h** Quantification of the *tbx1*:EGFP-positive ventricular cardiomyocytes compared to whole ventricle (in percentage) reveals no change over developmental period from 26 hpf ($n = 5$), 54 hpf ($n = 11$), to 72 hpf ($n = 15$). Each data point represents averaged percentage per heart; means ± s.d. **i–k** At 54 hpf, cardiomyocytes of the later-differentiated distal ventricle express *tbx1*:EGFP (arrowhead), as do MHC-negative progenitors of the OFT (asterisks). **l–n** The differentiated BA at 3 dpf is positive for *tbx1*:EGFP. Scale bars 200 μm (**d**, **e**), 10 μm (**f**, **g**, **i–n**)

speed imaging and reconstruction of the beating zebrafish heart from 28 to 52 hpf[48]. We imaged *tbx1:EGFP;myl7:DsRed2* embryos from a lateral view (right side) to optimally resolve the migrating and looping ventricle (Fig. 4a–c, Supplementary Movie 6). Linking to our previous time course, we observed the *tbx1:EGFP*-positive/*myl7:DsRed2*-negative cells connected to the differentiating, *myl7:DsRed2*-fluorescent myocardium at 28 hpf (Fig. 4a); displacement of these cells due to the beating differentiated myocardium supports their continuous incorporation into the heart tube (Supplementary Movie 7). Throughout ventricle looping, the initially solely *tbx1* reporter-positive cells became re-arranged within the ventricle, intercalated with *myl7:DsRed2*-expressing differentiated cardiomyocytes, and gradually turned on *myl7:DsRed2* expression (Fig. 4b, c), in agreement with the previously described late SHF-derived myocardial differentiation[22,24,27]. Imaging of still hearts showed that all ventricle-incorporated *tbx1:EGFP*-expressing cells were differentiated and expressed *myl7:DsRed2* by 54 hpf (Fig. 4d). In addition, *tbx1* reporter-expressing, undifferentiated cells that gradually upregulated *myl7* expression also appeared at the IFT (Fig. 4a–c).

In our time-lapse, we noted that from time point 35 onward (~39 hpf), a new cluster of *tbx1* reporter-expressing cells appeared at the OFT that expanded throughout the rest of recorded cardiac

development, but never crossed into the ventricle (Fig. 4b, c, Supplementary Movie 6); we hypothesized that these cells are BA progenitors. Indeed, we detected *tbx1:EGFP* expression in the differentiated BA smooth muscle at 3–4 dpf (Fig. 4e), consistent with our genetic lineage tracing (Fig. 2s–u). We corroborated our live-imaging data by examining *tbx1:EGFP* reporter expression in dissected hearts of transgenic embryos at 26, 54, and 72 hpf (Fig. 4f–n); quantification documented no overt change in the number of *tbx1:EGFP*-positive ventricular cardiomyocytes over the three observed time points (Fig. 4h). At 26 hpf, we observed *tbx1:EGFP*-positive cells at the distal ventricle of hearts that did not express MHC at this time (Fig. 4f, g), and at 54 hpf, we detected differentiated cardiomyocytes expressing *tbx1:EGFP* as well as potential BA precursors at the OFT that did not express MHC (Fig. 4i–k). At 3 dpf, the differentiated smooth muscle but not the endothelium of the BA was clearly marked by *tbx1:EGFP* expression (Fig. 4l–n, Supplementary Fig. 4).

Altogether, our live-imaging data confirmed by analysis in isolated hearts (i) provide real-time observation of late-differentiating ventricle myocardium, and (ii) confirm that *tbx1* reporter-positive cells that comigrate with the early-differentiating cardiac cone subsequently differentiate to ventricular cardiomyocytes, including the late-differentiating myocardial pool at the arterial pole[24,27]. Moreover, we detect a second

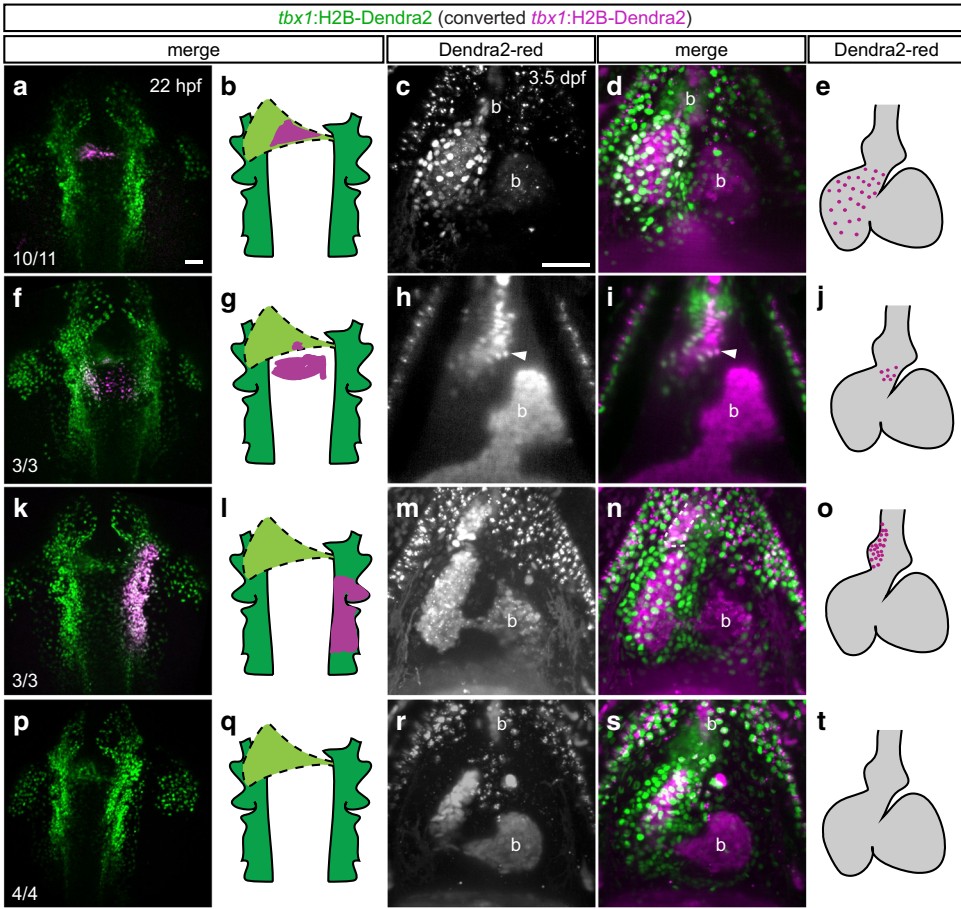

**Fig. 5** The *tbx1*+ cone forms the ventricular myocardium but not the BA. **a**, **f**, **k**, **p** Maximum intensity projections and schematics of representative photoconverted (**a**, **f**, **k**) or control (**p**) *tbx1*:Dendra2 embryos; dorsal views, anterior to the top. At 22 hpf, *tbx1*:Dendra2-expressing embryos were illuminated with a 405-nm laser in a confined region of interest to convert Dendra2-green to Dendra2-red in specific *tbx1* reporter-expressing domains. **c**, **h**, **m**, **r** SPIM-imaged hearts and graphical representations of embryos photoconverted as in **a**, **f**, **k** and **p** and stopped from contracting with BDM at 3.5 dpf; maximum intensity projections (**c**, **d**, **m**, **n**, **r**, **s**) or optical *Z*-section (**h**, **i**), ventral views, anterior to the top. **a–e** Dendra2-red-positive sheath cells give rise to ventricular cardiomyocytes, including the SHF-derived distal ventricle, but not to the BA. The red signal within the BA (**b**) derives from autofluorescent blood also detected in non-photoconverted *tbx1*:Dendra2 embryos (see **p–t**). **f–j** Medial migrating cells posterior to the cardiac cone contribute to the most distal myocardium at the dorsal side of the heart (arrowhead in **h**, **i**) and to a proximal portion of the BA in 3/3 analyzed embryos. **k–o** Photoconversion of a broad area in the *tbx1*:Dendra2-positive pharyngeal ALPM posterior to and on the right of the linear heart tube marks the right side of the BA (dotted outline in **n**). **p–t** Red signals in the chambers and on top of the pericardium are due to unspecific autofluorescence also detected in controls. Scale bars 50 μm

phase of *tbx1* reporter-expressing cells adding to the heart at later stages and potentially making the smooth muscle of the BA.

**The *tbx1* sheath forms ventricular myocardium**. While corresponding to our *tbx1:creERT2* lineage tracing (Fig. 2s–u), active EGFP reporter expression is not strict evidence for lineage association. To confirm the lineage contribution of the *tbx1* reporter-expressing cardiac sheath to the ventricular myocardium and OFT, we performed optogenetic lineage tracing using the transgenic line *Tg(−3.2tbx1:H2B-Dendra2)^{zh704}* (subsequently as *tbx1:Dendra2*); in *tbx1:Dendra2*, a nuclear histone 2B-linked Dendra2 fluorophore is constitutively green-fluorescent and turns irreversibly red upon photoconversion[49]. We photoconverted *tbx1*:Dendra2-positive cells of the cardiac cone and trailing sheath at 22 hpf (Fig. 5a, b) and detected Dendra2-red positive cardiomyocytes at 3.5 dpf throughout the ventricle, including the SHF-assigned distal portion and the inner curvature ($n = 10/11$, $N = 4$; *n*: total number of individual embryos analyzed, *N*: number of individual experiments performed; Fig. 5c–e, Supplementary

Fig. 5). These observations suggest that the *tbx1:Dendra2*-expressing sheath contributes to FHF- and SHF-linked ventricular myocytes. When converting *tbx1:Dendra2*-expressing cells at the base of and posterior to the cone in the area previously assigned to harbor additional myocardial SHF precursors[30] (Fig. 5f, g), we detected a few distal cardiomyocytes only on the dorsal side of the ventricle labeled by Dendra2-red ($n = 3/3$, $N = 1$; Fig. 5h–j). We identified the majority of the ventricle to be derived from *tbx1:Dendra2*-expressing cells present already in the cone and only a small fraction of cardiomyocytes to follow later. SHF contribution was previously reported to comprise 40–50% of the ventricular myocardium[21,22,24]; thus, our data suggest that the majority of late-differentiating cardiomyocyte progenitors enter the heart trailing the early-differentiating ventricular progenitors by 22 hpf and as part of a continuous *tbx1* reporter-expressing cell sheath.

In contrast to ventricular cardiomyocytes, we could not detect any Dendra2-red cells in the BA after photoconversion at the trailing end of the cardiac cone ($n = 0/11$, $N = 4$; Fig. 5a–e, Supplementary Fig. 5, red signal due to autofluorescence from

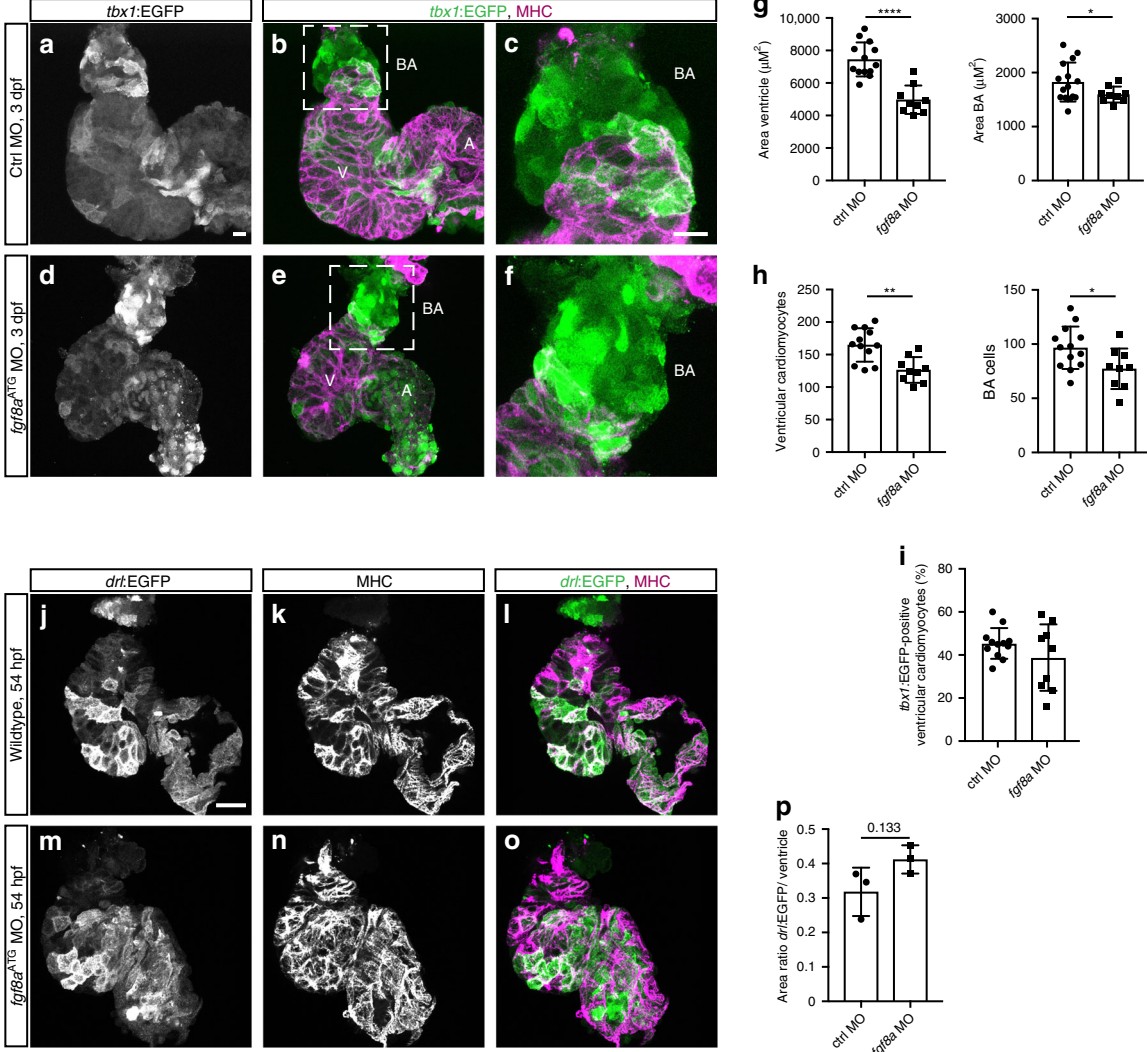

**Fig. 6** *fgf8a* knockdown leads to perturbed ventricle and BA formation. **a–f** Top-down 7-μm confocal section of wild-type control (**a–c**) and *fgf8a*^ATG morphant (**d–f**) hearts at 3 dpf. *tbx1*:EGFP, counterstained with anti-GFP labels the bulbus arteriosus (BA) and scattered ventricular (V) and atrial (A) cardiomyocytes, counterstained with anti-MHC. **c, f** depicts a magnification of the framed area in **b** and **e**. **g** Quantification of the ventricular (*n* = 13) and BA area (*n* = 15) in control morpholino (ctrl MO)-injected hearts compared to *fgf8a*^ATG morphant hearts (*n* = 9) reveals that both ventricle and BA are significantly smaller upon *fgf8a* knockdown (****P < 0.0001, *P = 0.0406). Each data point represents the averaged ventricular or BA area from one heart. **h** Quantification of the cell number of ventricular cardiomyocytes and the BA cells in ctrl MO hearts (*n* = 12) compared to *fgf8a*^ATG morphant hearts (*n* = 9) displays significantly less cells in both ventricle and BA upon loss of *fgf8a* (**P = 0.001, *P = 0.0305). Each data point represents the total number of nuclei per region from one heart. **i** Quantification of the *tbx1*:EGFP-positive ventricular cardiomyocytes compared to whole ventricle (in percentage) in ctrl MO hearts (*n* = 12) and *fgf8a*^ATG morphant hearts (*n* = 9) shows no difference (P = 0.2603). **j–o** Top-down 3-μm confocal sections of wild-type control (**j–l**) and *fgf8a*^ATG morphant (**m–o**) hearts at 54 hpf. *drl*:EGFP counterstained with anti-GFP labels FHF-derived cardiomyocytes co-marked by anti-MHC. **p** The ratio between the area of *drl*:EGFP-positive cells and the area of the entire ventricle is not significantly different comparing controls and *fgf8a*^ATG morphants (P = 0.133). Each data point presents the calculated ratio from one heart. Means ± s.d. ****P ≤ 0.0001, unpaired *t*-test with Welch correction. Scale bars 10 μm

blood, compare to Fig. 5k–n, p–t). In contrast, consistent with earlier position-based lineage tracing[23,29], we detected Dendra2-red-positive cells in the BA when converting the pharyngeal ALPM lateral to the forming cardiac cone (*n* = 15/15, *N* = 4; Fig. 5k–o). The right and left sides of the BA were exclusively formed from the corresponding side with no discernible crossover (*n* = 3/3 left side and *n* = 3/3 right side, total *n* = 6/6, *N* = 3; Fig. 5k–o, Supplementary Fig. 6). Moreover, different regions of the pharyngeal ALPM contributed to different parts of the BA on a proximal-to-distal axis and additionally labeled different craniofacial structures (*n* = 3/3 proximal part, *n* = 3/3 medial part, *n* = 3/3 distal part, *N* = 2; Fig. 5k–o; Supplementary Fig. 6).

Taken together, our data document that the majority of, if not all, ventricular cardiomyocytes stem from a *tbx1* reporter-expressing progenitor sheath that participates in cardiac ALPM fusion, contributes to the linear heart tube, and trails the prospective arterial pole. In contrast, SHF-assigned BA precursors within the pharyngeal ALPM add to the heart at a subsequent stage as more distantly trailing cells, as has been reported for OFT lineages in mammals[4,5].

**FGF signaling controls *tbx1*+ progenitor addition.** FGF signaling influences cardiac patterning, including SHF development in various chordates[50]. Correspondingly, zebrafish embryos

mutant for *fgf8a* (*acerebellar, ace*) or upon *fgf8a* morpholino knockdown form a severely hypoplastic ventricle with comparably normal atrium[27,51]. Moreover, chemical perturbations using the pan-FGF signaling inhibitor SU5402[52–54] from the earliest phases of heart formation result in reduced late-differentiating myocardium and loss of SHF marker expression at the arterial pole[22,27].

To determine the temporal requirement for FGF signaling on the different phases of cardiac contribution from the *tbx1* reporter-expressing ALPM field, we first revisited the *fgf8a* phenotype using the verified *fgf8a* translation-blocking morpholino MO3-*fgf8a*[ATG 55] (Fig. 6a–i). We measured ventricle and BA size upon *fgf8a* perturbation in isolated hearts and compared them to hearts of control morpholino-injected embryos: *fgf8a* morphants still

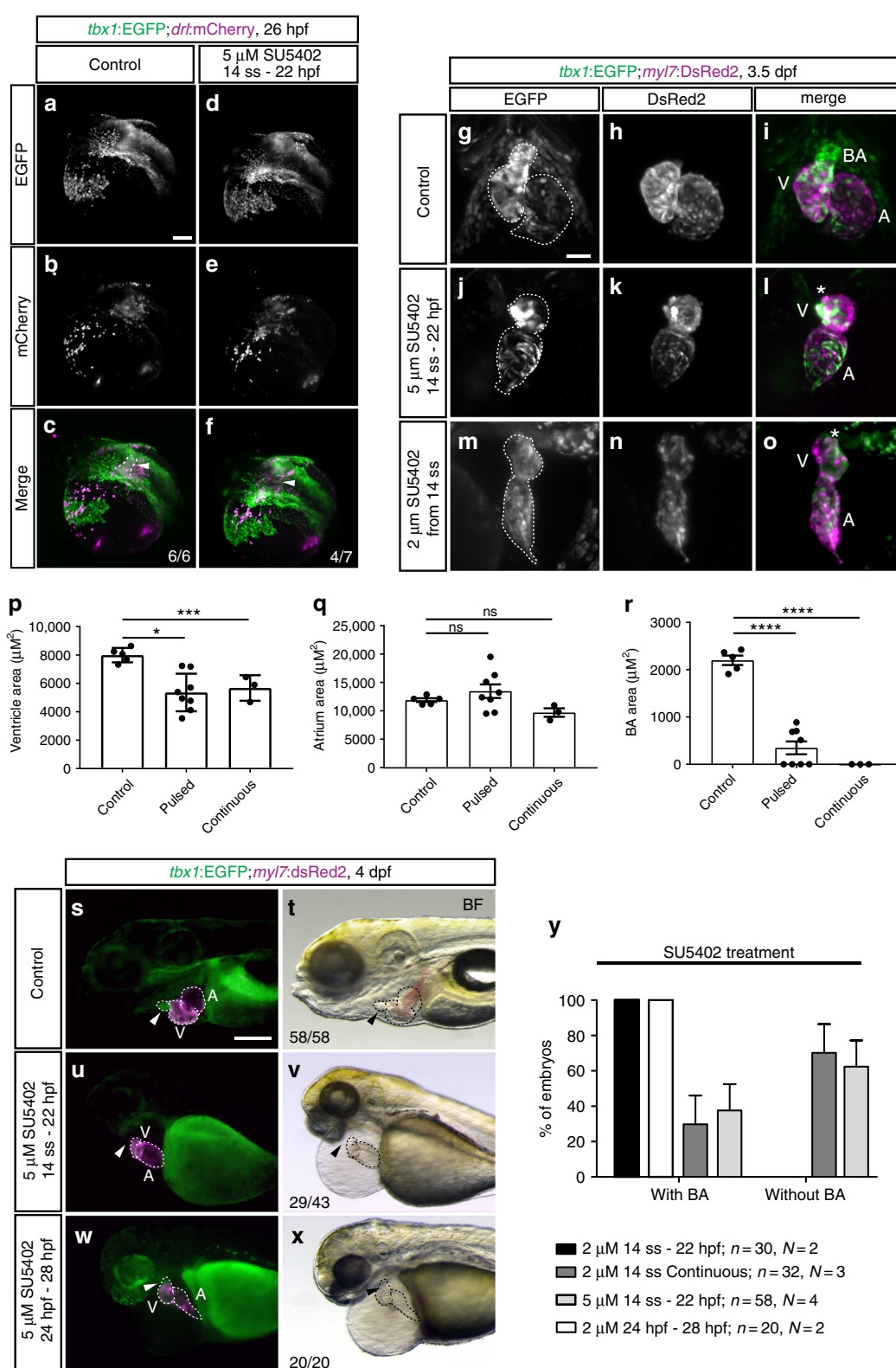

faithfully expressed *tbx1*:EGFP with no significant change of reporter-expressing cardiomyocyte number within the ventricle (Fig. 6i) and formed a BA (Fig. 6d–f), yet both ventricle and BA were significantly smaller with less cells (Fig. 6g, h) consistent with previous reports[27,51,54]. The impact on ventricle and BA size is unlikely from a strong disruption of early- vs. late-differentiating myocardium (FHF or SHF territories, respectively), as at 54-hpf expression of *drl*:EGFP that demarcates FHF-derived myocardium[21] appeared insignificantly affected upon *fgf8a* perturbation (Fig. 6j–p).

To resolve the temporal influence of FGF signaling, we treated *tbx1:EGFP* embryos with SU5402 when *tbx1* reporter-expressing ventricular progenitors migrate to contribute to the forming heart tube. When we initiated SU5402 treatment before medial migration at 14 ss and perturbed FGF signaling throughout sheath migration (pulse treatment from 14 ss to 22 hpf), 5 µM SU5402-treated embryos showed diminished but still ongoing migration of the *tbx1*:EGFP-expressing sheath by 26 hpf (Fig. 7a–f). At 3.5 dpf, the ventricle of these embryos contained *tbx1*:EGFP-expressing cardiomyocytes (Fig. 7g–l, Supplementary Movie 8). We observed similar results when continuously perturbing FGF signaling with a lower dose of 2 µM SU5402 from 14 ss onward till imaging at 3.5 dpf (Fig. 7m–o, Supplementary Movie 9). In line with previous findings[54], the ventricles of pulse (5 µM, 14 ss–22 hpf) or continuously (2 µM, from 14 ss to imaging at 3.5 dpf) SU5402-treated embryos were significantly smaller (Fig. 7p), while the size of the atrium was not significantly changed (Fig. 7q). These results demonstrate an impact of FGF signaling during *tbx1* reporter-expressing sheath migration, consistent with an early role of FGF signaling in ventricle formation.

In contrast to the mere reduction in ventricle size upon FGF inhibition (Fig. 7g–p), we observed a more striking effect on BA addition when perturbing FGF signaling during *tbx1* reporter-expressing sheath migration (Fig. 7r). To better resolve the temporal requirement of FGF signaling in this process, we treated *tbx1* reporter-expressing or DAR-4M-labeled embryos with an early (14 ss–22 hpf) or late (24 hpf–28/34 hpf) pulse of SU5402 (Fig. 7s–x; Supplementary Fig. 7). Even though BA addition takes place after primary heart tube formation at 24 hpf, BA formation was completely absent when assessed at 3.5 dpf in a substantial number of embryos pulse-treated between 14 ss and 22 hpf with 5 µM SU5402 (Fig. 7u, v, y; Supplementary Fig. 7). While pulsing SU5402 treatment between 14 ss and 22 hpf with 2 µM SU5402 invariantly allowed for detectable BA differentiation, the majority of embryos continuously treated from 14 ss onward with 2 µM SU5402 failed to form a BA (Fig. 7u, v, y). In contrast, SU5402 pulse treatment after heart tube formation from 24 hpf to 28 hpf

or to 34 hpf merely caused formation of smaller BA and never caused loss of the differentiated BA, as confirmed by DAR-4M; this effect was also dependent on the SU5402 concentration (Fig. 7w–y, Supplementary Fig. 7).

Taken together, our data determine a temporally defined requirement for FGF signaling during formation and migration of the ventricle progenitor sheath. Further, our findings support a model in which differential levels and timing of FGF control the continuous addition of ventricle and BA progenitors to the heart tube, with a sensitive window for BA determination before its progenitors leave the pharyngeal ALPM.

## Discussion

The concept of distinct phases of differentiating cell types contributing to individual parts of the multi-chambered heart is deeply rooted in chordate evolution. Here, we visualized the formation of the zebrafish ventricle and OFT structures from a continuously differentiating progenitor sheath that emerges from *tbx1* reporter-expressing ALPM and surrounds the emerging endocardium. Our data connect, are consistent with, and extend previous end-point analyses of early- vs. late-differentiating myocardium and the formation of OFT structures including the BA. We further reveal a temporal influence and sensitivity of FGF signaling on ventricle myocardium vs. BA smooth-muscle formation in zebrafish. Our work emphasizes cardiac development as part of a *tbx1*-expressing cardiopharyngeal progenitor field within the bilateral ALPM that is already significantly pre-patterned before its medial migration forms the heart.

Reporter expression and lineage tracing (Figs. 1, 2, Supplementary Figs. 2, 3) established the isolated zebrafish *tbx1 cis*-regulatory region as a putative marker of the CPF that resides within the ALPM aside cranial endoderm and neural crest lineages. We have combined different strategies to demonstrate that our *tbx1* regulatory elements visualize early- as well as late-differentiating cardiac lineages. When live-imaging *tbx1*:EGFP-expressing embryos from stages before heart field migration (14 ss) up to cardiac looping stages (54 hpf), we robustly detected *tbx1* reporter-expressing cells at the arterial pole of the ventricle that were connected to the differentiated *myl7*-expressing myocardium and started moving in sync with heartbeat of the primitive heart tube (Figs. 3, 4, Supplementary Movies 3–7); this observation suggests that undifferentiated *tbx1* reporter-expressing cells are already physically linked to beating cardiomyocytes. These *tbx1*:EGFP-expressing progenitors then gradually upregulated *myl7* expression, confirming their identity as cardiomyocyte progenitors equivalent to late-differentiating SHF-liked cells (Fig. 4a–c, Supplementary Movie 6)[26,27]. Independent of

**Fig. 7** FGF signaling differentially affects *tbx1+* ventricular and BA precursors. **a–f** Maximum intensity projections of representative *tbx1*:EGFP;*drl*:mCherry DMSO-treated controls or embryos treated with 5 µM SU5402 from 14 ss to 22 hpf; lateral/dorsal view, anterior to the left. FGF signaling-perturbed embryos show a defect in the *tbx1*:EGFP-expressing sheath (arrowhead) at the base of the forming heart tube (outline). **g–o** Maximum intensity projections of representative hearts of *tbx1*:EGFP;*myl7*:DsRed2 embryos, DMSO-treated controls (**g–i**, *n* = 5), treated with 5 µM SU5402 between 14 ss and 24 hpf (**j–l**, *n* = 8), or treated with 2 µM SU5402 continuously from 14 ss (**m–o**, *n* = 3); ventral views, anterior to the top, outlines mark the heart. FGF-perturbed embryos retain normal contribution of *tbx1*:EGFP-expressing cardiomyocytes to the ventricle upon pulsed or continuous signaling inhibition; ventricle (V), atrium (A), bulbus arteriosus (BA), and asterisks mark the missing BA upon SU5402 treatments. **p–r** Quantified ventricle, atrium, and BA area size in SU5402-treated embryos as in **g–o**. Pulsed (see **j–l**) or continuous (see **m–o**) FGF signaling inhibition diminishes ventricle size (**p**) and severely reduced to abolished addition of the *tbx1* reporter-expressing BA (**r**). The atrium is not significantly affected (**q**). Means ± SEM. ns $P > 0.05$, *$P \leq 0.05$, **$P \leq 0.01$, ***$P \leq 0.001$, and ****$P \leq 0.0001$, unpaired *t*-test with Welch correction. **s–y** SU5402 treatments affect BA development in time- and concentration-dependent manner; *n* indicates the number of embryos analyzed per condition, *N* indicates the number of experiments performed. **s–x** *tbx1*:EGFP;*myl7*:DsRed transgenic controls and embryos treated with DMSO or 5 µM SU5402 during (14 ss–22 hpf) or after (24 hp–28 hpf) heart tube formation; lateral views, anterior to the left. Absent BA formation can only be observed in embryos treated with SU5402 from mid-somitogenesis to heart tube stages (arrowhead in **u**, **v**, compare to **s**, **t**), but not when signaling inhibition is initiated at 24 hpf (arrowhead in **w**, **x**). **y** Quantification of the concentration-dependent effect on BA formation in FGF signaling-perturbed *tbx1*:EGFP;*myl7*:Red or DAR-4M-stained *myl7*:EGFP transgenics (see Supplementary Fig. 7). For 5 µM SU5402 from 14 ss to 22 hpf, no BA in *n* = 29/43 as assessed by *tbx1* reporter and *n* = 5/15 by DAR-4M, total *n* = 34/58, *N* = 4; DMSO-treated controls: normal BA in *n* = 58/58 by *tbx1* reporter and *n* = 10/10 by DAR-4M, total *n* = 68/68, *N* = 4. Scale bars 50 µm (**g**, **o**), 100 µm (**a–f**, **s–x**)

continued *tbx1* reporter activity, genetic and optogenetic lineage tracking supports the notion that most, and likely all, FHF- and SHF-assigned ventricular cardiomyocytes are contained within the *tbx1* reporter-expressing sheath that contributes to the cardiac cone and trails into the bilateral ALPM (Fig. 5a–j, Supplementary Fig. 5). Of note, the SHF in mouse has been characterized as epithelial sheet that undergoes epithelial-to-mesenchymal transitions and tension changes during its addition to the heart tube[7]. Our observations hint at similar dynamics of SHF cells in teleosts.

The observed addition of ventricle progenitors as a continuous process connects previously observed key time points of SHF-assigned, late-differentiating myocardium addition to the zebrafish heart tube[21,22,24,29]. Nonetheless, the position of SHF-assigned cells during heart tube assembly had previously remained ambiguous. *nkx2.5*:Kaede reporter-based optogenetic lineage tracking showed that most if not all ventricular myocardium is already condensed at the cardiac disk, but whether these cells then migrate with the emerging primitive heart tube or stay behind had remained unresolved[29]. A distinct group of *nkx2.5*:Kaede-expressing cells were found more posterior, seemingly outside of the forming heart tube, and shown to form a small portion of the distal ventricle and OFT myocardium[30]. Our *tbx1* reporter transgenics now consolidate these data points as parts of the continuous cell sheath forming the ventricle.

On the contrary, smooth-muscle cells in the BA contributed from the still bilateral pharyngeal ALPM lateral and posterior to the developing heart tube (Fig. 5k–o, Supplementary Fig. 6), a development that also appears in our live-imaging data (from ~39 hpf, time point 35 in Fig. 4b, c, Supplementary Movie 6). Inferred by reporter expression and not strictly lineage data, our analysis of dissected hearts at 54 hpf (Fig. 4f–n) confirms the existence of a *tbx1* reporter-expressing, non-myocardial cell population at the OFT pole, the position where we later find the differentiated BA (Fig. 4e, l–n). Addition of BA precursors to the distal ventricle has been reported as early as 48 hpf by 4,5-diaminofluorescein diacetate (DAF-2DA) staining that senses nitric oxide accumulating in the BA and likely requires functional maturation of smooth muscle[13]. A concise working definition (proposed in Fig. 1a) to distinguish between distal ventricle myocardium, collagenous OFT myocardium of the CA, and BA could further consolidate the various reports of distinct waves of SHF-assigned cells to the heart tube that have been referred to by mixed nomenclature[12]. Our data establish the accrual of ventricle myocardium and the subsequent addition of smooth-muscle progenitors as distinct phases of a continuous process.

FGF signaling has been previously reported to govern progenitor addition to the arterial pole of the zebrafish heart[22,27]. Consistent with these previous findings, we observed a diminished ventricle upon pan-FGF signaling perturbation during *tbx1* reporter-expressing sheath formation (Fig. 7g–o). We further detected absent BA formation when initiating chemical FGF signaling inhibition already before cardiac cone formation (Fig. 7h–l). In contrast, knockdown of the key cardiac FGF ligand gene *fgf8a* merely caused a smaller ventricle and BA (Fig. 6a–h), akin to pulsed exposure to a low dose of SU5402 during ventricle formation and BA accrual at 14 ss–22 hpf (Fig. 7f–h). To our knowledge, the drastic temporal influence of FGF signaling on BA formation has not been previously described in zebrafish. To exclude that the observed loss of the BA was not caused by edema formation that could potentially induce apoptosis due to extensive stretching forces, we analyzed BA formation in *tbx5a* morphants exhibiting the heartstring phenotype: in all analyzed *tbx5a* morphants that develop the reported heartstring phenotype (as well as milder forms of looping defects in *tbx5a* morphants)[56], the BA was properly formed and could be observed up to 5 dpf (Supplementary Fig. 7). As the heartstring phenotype is an extreme manifestation of a stretched heart and extensive edema, the loss of the BA in FGF signaling-perturbed embryos is not likely to occur due to apoptosis caused by the observed edema. FGF8 depletion in mouse results in aberrant OFT development, including severe aortic arch defects, a structure probably most comparable to the teleost BA[57,58]. We only observed complete failure of BA formation when all FGF signaling was blocked before cardiac cone stages, but not when perturbed later during heart tube stages or by sub-penetrant doses of SU5402 (Fig. 7r–l).

Our data are consistent with two effects of FGF signaling on zebrafish heart development after initial cardiac specification: first, FGF signaling regulates ventricular myocardium formation during medial migration of ventricular progenitors and cardiac cone formation; second, FGF signaling controls smooth-muscle precursors residing in the pharyngeal ALPM prior to their OFT addition. Work on chick SHF explants has established that loss of FGF signaling blocks proliferation and causes myocardial differentiation, while elevating FGF signaling drives SHF cells into smooth-muscle fates[59]. In *Ciona*, an FGF-driven regulatory circuit controls key cardiopharyngeal transcription factors including Tbx1/10 and regulates cell-cycle dynamics to permit differentiation of individual cardiac lineages[10,60,61]. These results are in line with our temporal and dose requirement for FGF during zebrafish ventricle and OFT formation (Fig. 7), suggesting that BA progenitors already have an assigned fate during heart cone formation. Regulation of OFT progenitors while they reside in the pharyngeal LPM points toward an FGF activity gradient mediated by adjacent structures.

Of note, we also detected a late-differentiating cardiomyocyte population at the venous pole, the IFT, of the heart (Fig. 4a–c), in accordance with previous findings of late IFT myocardial differentiation[27]. *tbx1* reporter-expressing cells at the IFT concomitantly expressed Isl1, confirming their SHF signature (Fig. 2i–p). We did not detect any Isl1 expression at the OFT, in accordance with other studies in zebrafish implicating Isl1 in IFT development[25,27,46]. We detected Isl1 + IFT cells that seemingly did not express the *tbx1* reporter at the time point of analysis (~22.7% of all Isl1-expressing IFT cells; Fig. 2q, Supplementary Movie 2). Whether this result is a consequence of dynamic *tbx1*: EGFP reporter expression or is indicative of late-differentiating IFT subpopulations with distinctive gene-expression signatures deserves a more detailed analysis. While we here focused on the development of the arterial pole, these observations warrant application of our *tbx1* reporter transgenics for the elucidation of zebrafish IFT formation.

Altogether, our data provide new insights into the dynamics of ventricle and OFT formation and integrate their mechanistic separation as distinct phases of a continuous developmental process in zebrafish.

## Methods

**Animal husbandry.** Zebrafish (*Danio rerio*) were maintained, collected, and staged principally as described[62] and in agreement with procedures mandated by the veterinary office of UZH and the Canton of Zürich, or in accordance with the guidelines of the Max Delbrück Center for Molecular Medicine and the local authority for animal protection (Landesamt für Gesundheit und Soziales, Berlin, Germany) for the use of laboratory animals, and following the "Principles of Laboratory Animal Care" (NIH publication no. 86–23, revised 1985) as well as the current version of German Law on the Protection of Animals. Embryos were raised in temperature-controlled incubators without light cycle at 28 °C unless specified differently in the text.

**Vectors and transgenic lines.** All transgenic lines newly generated in this work have been assigned unique ZFIN designations. The upstream *cis*-regulatory region of the zebrafish *tbx1* gene (ZDB-GENE-030805-5) was amplified from zebrafish wild-type genomic DNA with primers 5′-GCTTATACGCACGACTGC-3′ (forward) and 5′-TGTGTCGATCGCGTATCGC-3′ (reverse) with the Expand Hi-Fidelity PCR kit (Roche). The 3242-bp upstream region of *tbx1* was TOPO-cloned into the

pENTR™ 5′-TOPO® TA Cloning® plasmid (Cat#59120; Invitrogen) according to the manufacturer's instructions to obtain pAF006 (pENTR/5′_tbx1).

Subsequent cloning reactions were performed with the Multisite Gateway system with LR Clonase II Plus (Cat#12538120; Life Technologies) according to the manufacturer's instructions.

tbx1:EGFP (pAF008 or pDestTol2pA2_tbx1:EGFP) and tbx1:H2B-Dendra2 (pAF048 or pDestTol2pA2_tbx1:H2B-Dendra2) were assembled from pAF006 together with Tol2kit #383 (pME-EGFP) or pKP003 (pME-H2B-Dendra2, cloned from p55-H2B-Dendra2[63]), #302 (p3E_SV40polyA), and #394 (pDestTol2A2) as backbone[64]. pAF008 was used to generate transgenic strain Tg(−3.2tbx1:EGFP)$^{zh702}$ based on founder line I and the additional lines for comparison, as depicted in Supplementary Fig. 1 to ensure faithful transgene expression. pAF048 was used to generate transgenic strain Tg(−3.2tbx1:H2B-Dendra2)$^{zh704}$, one of two lines that showed basically identical expression and optogenetic properties (see also text for details and Supplementary Fig. 5).

We cloned tbx1:creERT2 (pAF038 or pDestTol2CY_tbx1:creERT2,alpha-crystallin:Venus) by combining pAF006 with pCM293 (pENTR/D_creERT2)[65], Tol2kit vector #302[64], and pCM326 (pDestTol2CY, containing the alpha-crystallin: Venus cassette as transgenesis marker) as backbone[21]. This vector was used to generate transgenic strain Tg(−3.2tbx1:creERT2,cryaa:Venus)$^{zh703}$, which we selected as the best line after screening several transmitting founders (see also text for details).

The transgene hsp70l:Switch (pAF040 or pDestTol2CY_hsp70l:loxP-STOP-loxP-EGFP,alpha-crystallin:Venus) was assembled from pDH083[66] by transfer of the loxP cassette into pENTR5′ (generating pENTR/5′_hsp70l:loxP-STOP-loxP), Tol2kit #383 and #302[64], and pCM326 as backbone. This vector was used to generate the transgenic strain with ZFIN designation Tg(−1.5hsp70l:loxP-STOP-loxP-EGFP, cryaa:Venus)$^{zh701}$.

For Tol2-mediated zebrafish transgenesis, 25 ng/µL Tol2 mRNA were injected with 25 ng/µL plasmid DNA[64,67]. F0 founders were screened for specific EGFP or alpha-crystallin:YFP expression, raised to adulthood, and screened for germline transmission. Single-insertion transgenic strains were established and verified through screening for a 50% germline transmission rate in outcrosses in the subsequent generations as per our previously outlined procedures[67].

For tbx1:EGFP, lines with more than one insertion were not followed up (lines III and VI, see Supplementary Table 1) and of the remaining four lines, three were retained: lines I (Tg(−3.2tbx1:EGFP_I)$^{zh702}$), IV, and V. Due to its clean expression pattern without unspecific signals, experiments were performed in line I in generation F2 or beyond (Supplementary Table 1, Supplementary Fig. 1), and some experimental outcomes confirmed in the other retained lines. For tbx1:creERT, two lines were isolated and retained. The main experiments were performed in line II (Fig. 2s–u, Supplementary Fig. 2) due to higher and more stable switching mosaicism observed in the heart compared to line I (Supplementary Fig. 3). For tbx1:Dendra2, two lines were isolated and retained, line I and line II; all experiments were performed in line II due to the unclean insertion pattern observed in line I (Supplementary Fig. 5).

Additional already-established transgenic lines used in this study included drl: mCherry (Tg(−6.3drl:mCherry)$^{zh705}$)[68], myl7:DsRed2[69], ubi:Switch[65], and drl: creERT2[21].

**Morpholino injections**. The previously characterized and validated[55,56] fgf8a$^{ATG}$ and tbx5a$^{ATG}$ morpholinos (MO3-fgf8a$^{ATG}$: 5′-GAGTCTCATGTTTA-TAGCCTCAGTA-3′; ZFIN ID: ZDB-MRPHLNO-050714-1, MO2-tbx5a$^{ATG}$: 5′-CCTGTACGATGTCTACCGTGAGGC-3′; ZFIN ID: ZDB-MRPHLNO-060328-3), as well as the standard control morpholino (5′-CCTCTTACCTCAGTTA-CAATTTATA-3′) were obtained by Gene Tools, LLC and injected in the yolk of one- to four-cell-stage embryos.

**Agarose sections**. Transverse sections of agarose-embedded embryos at 3 dpf were performed essentially as previously described[47]: embryos were fixed in 4% paraformaldehyde (PFA), embedded in 6% low-melting-point agarose (Sigma), cut into 130-mm vibratome sections (VT1000S, Leica), and mounted with DAPI-containing Vectashield (Cat#H-1200; Vector Laboratories).

**Whole-mount in situ hybridization**. First-strand complementary DNA (cDNA) was generated from wild-type zebrafish RNA isolated with Superscript III First-Strand Synthesis kit (Cat#18080051; Invitrogen). DNA templates were generated using first-strand cDNA as PCR template and the following primers: EGFP with 5′-ATGGTGAGCAAGGGCGAGGAGC-3′ (forward) and 5′-TAAATACGACTCACTA-TAGGG-3′(reverse); tbx1 with 5′-TATTCCGGATCCAACTCAGC-3′ (forward) and 5′-TTATCTGGGTCCGTAGTC-3′ (reverse). For in vitro transcription, the T7 RNA polymerase promoter 5′-TAATACGACTCACTATAGGG-3′ was added to the 5′-end of reverse primers. In situ hybridization probes were made by in vitro transcription using T7 RNA polymerase and DIG-labeled NTPs (Cat#11277073910; Roche). RNA was precipitated with lithium chloride in ethanol and dissolved in DEPC water. Embryos were fixed in 4% PFA overnight at 4 °C, transferred into 100% methanol, and stored at −20 °C until in situ hybridization. In situ hybridization of whole-mount zebrafish embryos was performed according to published protocols[70].

**Antibody staining**. Embryos were fixed in 4% formaldehyde, 0.1% TritonX in PEM (0.1 M PIPES, 2 mM MgSO$_4$, and 1 mM EDTA) for 2–4 h at room temperature, washed in 0.1% PBS TritonX (PBSTx), and permeabilized in 0.5% PBSTx. Hearts from 26-, 54-, and 72-hpf zebrafish embryos were dissected in Tyrode's solution (136 mM NaCl, 5.4 mM KCl, 1 mM MgCl$_2$ × 6H$_2$O, 5 mM D (+)glucose, 10 mM HEPES, 0.3 mM Na$_2$HPO$_4$ × 2H$_2$O, and 1.8 mM CaCl$_2$ × 2H$_2$O; pH 7.4) with 20 mg/mL BSA and fixed with Shandon™ Glyo-Fixx™ (Cat#9990920; Thermo Fisher Scientific™) for 20 min at room temperature. Blocking was done in blocking buffer containing 5% goat serum, 5% BSA, 20 mM MgCl$_2$ in PBS, and embryos/ hearts incubated with primary antibodies diluted in blocking buffer at 4 °C overnight. Primary antibodies used were anti-MHC (MF20 supernatant, DSHB, 1:50), anti-GFP (Abcam, ab13970, 1:500 or Sigma, G1544, 1:100), and anti-Isl1 (GeneTex, GTX128201, 1:50). Alexa-conjugated secondary antibodies (A11039, A11004, A11008, and A11012 ThermoFisher Scientific) were added at 1:500 in 0.1% PBSTx at 4 °C overnight. Embryos were washed several times in 0.1% PBSTx. DAPI-containing Vectashield was added and embryos were kept in the mounting medium until imaging. Before imaging, embryos were mounted in 1% low-melting-point agarose. Dissected hearts were washed overnight in blocking buffer and mounted in the ProLong Gold antifade reagent with DAPI (Cat#P36935; ThermoFisher Scientific™).

**CreERT2-based lineage tracing**. Lineage-tracing experiments were performed by crossing female hsp70l:Switch or ubi:Switch reporter carriers with male creERT2 driver transgenics[67]. Embryos were induced using 4-OHT (Cat#H7904: Sigma) from fresh and/or preheated (65 °C for 10 min) stock solutions in DMSO with a final concentration of 10 µM in E3 embryo medium as per our established protocols[47]. Heat-shocks were performed for 60 min at 37 °C in glass tubes in a water bath.

**Microscopy and image analysis**. Stereomicroscopy images were obtained on a Leica M205FA equipped with a Leica DFC450C digital camera. Confocal images of transverse sections were obtained on an inverted Zeiss LSM710 confocal microscope with a Plan-Apochromat ×40/1.3 Oil DIC M27 objective. Confocal imaging of whole-mounts and dissected hearts was done with a Leica SP8 upright confocal microscope using a HC PL APO ×20/0.5 water objective and Leica SP8 inverted confocal microscope using a HC PL APO CS2 ×63 glycerol/NA 1.3 objective, respectively.

SPIM/lightsheet microscopy was performed on a Zeiss Z.1. Embryos were embedded in 1% low-melting-point agarose and 0.016% ethyl 3-aminobenzoate methanesulfonate salt (Tricaine, Cat#A5040; Sigma) in E3 embryo medium in a 50-µL glass capillary. Heartbeat was stopped with 30 mM 2,3-butanedione monoxime (BDM, Cat#B0753; Sigma) as indicated in individual experiments.

Dendra2 photoconversion experiments were performed on an inverted Zeiss LSM710 confocal microscope with the Plan-Apochromat ×20/0.8 M27 or LD LCI Plan-Apochromat ×25/0.8 Imm Korr DIC M27 objectives. Embryos were embedded in 1% low-melting-point agarose and 0.016% tricaine in E3 embryo medium in glass-bottom plates orienting the anterior dorsal side of the embryo toward the bottom of the plate.

Panoramic SPIM and high-resolution SPIM of the beating zebrafish heart, as well as image processing (Mercator projections and reconstruction of the beating heart) were performed essentially as described[44,48].

Images were processed using Leica LAS, ImageJ/Fiji, Imaris, and Photoshop CS6. The area of the BA, ventricle, and regions positive for drl:EGFP was measured from confocal image Z-projections of dissected hearts using ImageJ/Fiji.

**Chemical treatments**. SU5402 (Sigma) was administered to embryos at concentrations ranging from 2 to 5 µM in E3 at desired stages and for specific periods of time as indicated in the text. The drug was washed out through several washing steps with E3. Diaminorhodamine-4M AM solution (DAR-4M, Cat#D9194; Sigma) was diluted 1:1000 in E3 containing 0.003% 1-phenyl-2-thiourea (PTU, Cat#P7629; Sigma) and live embryos were incubated in the staining solution for at least 48 h at 28 °C. DAR-4M was washed out through several washing steps before imaging. Controls were treated with equivalent amounts of DMSO.

**Statistical analysis**. Statistical tests were performed with GraphPad Prism 7.04. All tests were performed as two-tailed, unpaired t-tests with Welch correction. Statistical significance was determined by a P-value ≤ 0.05 (ns P > 0.05, *P ≤ 0.05, **P ≤ 0.01, ***P ≤ 0.001, and ****P ≤ 0.0001).

**Data availability**. The authors declare that all data supporting the findings of this study are available within the article and its Supplementary Information files or from the corresponding author upon reasonable request.

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

## Acknowledgements

We thank Sibylle Burger and Seraina Bötschi for technical and zebrafish husbandry support, Dr. Stephan Neuhauss and his lab for zebrafish support, the Center for Microscopy and Image Analysis (ZBM) at UZH for imaging support and equipment, Hanyu Qin and Dr. Periklis Pantazis for optogenetics advice, and Dr. Deborah Yelon for critical input to the manuscript. This work has been supported by a Swiss National Science Foundation (SNSF) professorship [PP00P3_139093] and SNSF R'Equip grant 150838 (Lightsheet Fluorescence Microscopy), a Marie Curie Career Integration Grant from the European Commission [CIG PCIG14-GA-2013-631984], the Canton of Zürich, the UZH Foundation for Research in Science and the Humanities, and the Swiss Heart Foundation to C.M.; the Helmholtz Young Investigator Program VH-NG-736, Deutsche Forschungsgemeinschaft (DFG) PA2619/1-1, and Marie Curie Career Integration Grant from the European Commission (WNT/CALCIUM IN HEART-322189) to D.P.; Company of Biologists and EuFishBioMed travel grants to K.D.P.

## Author contributions

A.F., K.D.P., A.M.M., D.P., and C.M. designed, performed, and analyzed the experiments; K.D.P. performed panoramic SPIM under the guidance of J.H.; M.M. performed and analyzed the experiments in Fig. 4a–e and associated movies as supervised by J.H.; E.C.B. performed lineage trace experiments in Supplementary Fig. 2; D.P. and C.M. supervised the project; and A.F., D.P., and C.M. compiled the data and wrote the manuscript.

## Additional information

**Competing interests:** The authors declare no competing interests.

