## [Peer Review File · Nature Communications]

Reviewers' Comments:

Reviewer #1:

Remarks to the Author:

This is a very interesting paper that makes significant contributions to our knowledge of early heart development. In particular, it provides new insights and clarifications of previous models or hypothesis on how cells are added to the developing heart, with a focus on the ventricle and bulbus arteriosus (BA)/outflow tract. Utilizing transgenics, optigenetics and high-speed imaging, the authors identify an undifferentiated sheath of *tbx1* reporter-expressing cells that are continuously added to, and gradually differentiate at, the arterial pole. In addition, the authors find BA progenitors reside in the *tbx1* reporter-positive pharyngeal ALPM and migrate later to contribute to the late-differentiating distal pole of the ventricle to become smooth muscle. Using both FGF8 morpholinos and pharmacological inhibitors of FGF receptors, they identified a distinct temporal requirement for FGF signaling in controlling ventricle size and BA specification. Overall, this is a very strong contribution to the field, and helps pull together a wide range of previous observations into a consistent model for heart development, and helps compare heart development in zebrafish to mammals. A few points need clarification.

Major concerns

1. Cell quantification is lacking from most of the fate-mapping imaging analysis. This should be possible for at least Figures 2 and 4. In the analysis of co-expression of SHF marker *Isl1* and the *tbx1* transgene (Fig 2), it appears that very few cells co-express, a majority are exclusive for either *Isl1* or *tbx1* transgene. This suggests most of the SHF is not derived from *tbx1*⁺ cells. It would help to have a quantification from the images of how many cells (by counting) express just one or both markers. Similarly, in contrast to marking by *drl*-transgene (Fig 2) and *myl7*-transgene (Fig 4), what percentage/how many of the ventricle and atrial cardiomyocytes, endocardial cells, etc (Fig 2 and Fig 4) express the *tbx1* transgene?
2. In *fgf8a* morpholino treatment, the bulbus arteriosus (BA) and ventricle are smaller, as measured by area in the images. First, is this reduction in size due to a decreased number of cells (needs quantification) and/or size of cells? Second, is it a preferential loss of *tbx1* expressing cells or non-expressing cells? In the image (Fig 6), appears that the number of *tbx*-transgene positive cells are the same, at least in the BA. If so, what does this have to do with the relationship of FGF function and *tbx*-expressing lineages?
3. "In contrast to the mere reduction in ventricle size upon FGF inhibition, we observed a more striking and concentration-dependent effect on BA addition". This conclusion is somewhat questionable for BA development, and needs some additional analysis. It is also challenging when there is extensive edema; cells can get so stretched that they eventually die. An apoptosis assay would address that. Is it possible that the BA cells are there, and differentiated, but strong inhibition of FGF signaling prevents the expression of the *tbx*-transgene? Alternatively, this could be remedied by confocal microscopy of fixed embryo the hearts "in situ" in order to clarify.

Minor points

4. There are some concerns about the efficacy of CreERT2. Do you see mosaic activity that suggests inefficient activity?
5. In figure legend, mention what is insert in 1C through 1F
6. In supplemental Fig 2, for 3/3 or 4/4, it is not clear whether those different sections of the same embryo or different embryos. The level of the section (anterior-posterior axis) seems to be different, suggesting that they represent different sections of the same embryo. Also, in the heat shock control, it would help to put the exact same slice where there is a positive mark in their treated sample.

7. Merged of sup figure 4 E'' appears to be inverted (mirror image).

8. Indicate number of embryos analyzed by SPIM videos

Reviewer #2:

Remarks to the Author:

Felker et al. use *tbx1* transgenic reporter and lineage tracing tools to image the progressive formation of heart development with a focus on the accretion of cells to form the outflow tract (bulbus arteriosus, or BA). While not fundamentally impacting the current understanding of SHF contributions to the primitive heart tube, they provide some striking images and a useful contribution to describing how *tbx1*+ cardiopharyngeal progenitors contribute to the OFT. Chiefly, they describe a "continuous progenitor sheaf" that was not fully appreciated without live imaging. Consistent with recent observations, they document nicely the contribution of pharyngeal ALPM progenitors to the OFT and demonstrate a function for FGF signaling, also consistent with previous results. The study provides a very nice documentation of progressive OFT development and will be appreciated by the field. A few minor issues should be clarified.

1) Several transgenic lines were generated but it was unclear which of these were used in the study. Were results validated in more than one transgenic reporter? Were more than one CreERT line and DENDRA line also generated and did they recapitulate the normal pattern of *tbx1* expression? In particular, the CreERT line appears to label strongly the entire gut. Is this expected?

2) Did the lineage tracing of *tbx1* pharyngeal ALPM also label head mesoderm?

3) The temporal role of FGF signaling is unclear. BA appears to be often absent when SU was added as a pulse from 13-22 hpf. Yet the FGF8a morphants only have a reduced size of BA, even though it should be functioning through this same time frame. Is it just a hypomorph? Pulsing with SU from 24-34 hpf looks more like the morphant, with a smaller BA. How do the authors interpret this result? FGF is no longer required for the accretion of cells from the pharyngeal arches? Is reporter expression altered in PA mesoderm in the 13-22 hpf pulse?

Reviewer #3:

Remarks to the Author:

The work by Felker et al. examines the formation of the zebrafish ventricle and OFT structures by SPIM imaging with genetic and optogenetic lineage tracing, and by chemical and genetic perturbations. They show that the zebrafish ventricle forms through continuous addition from an undifferentiated *tbx1* reporter-expressing cell sheath followed by late phase accrual of the bulbus arteriosus. Furthermore, the authors report the temporal requirement for FGF signaling in controlling ventricle and BA size, and BA formation.

Major issues that should be addressed:

1. The selection criteria used to retain lines I, IV and V from all six transgenic lines should be mentioned. Preferentially in the Methods section.

2. The authors should show a corresponding control image for every section shown in Supplementary Figure 2A-E and 2G-K to appreciate the specificity of the GFP signal. Was the autofluorescence signal in skin and blood also present in heat shock control sections?

3. The authors should include higher magnification images of the heart area shown in Figure 2A-

A'''.

4. The Dendra2-red positive signal in Figure 5 and Supplementary Figure 6 is not always convincing as the intensity is similar to the autofluorescence signal in blood (e.g., Figure 5D and 5F and Supplementary Figure 6F and 6H). Images from unconverted control embryos captured and processed identical as the converted embryos should be shown side-by-side with the converted images at similar magnification. This will facilitate the interpretation of positive Dendra2-red signal.

5. A proper control group (five mismatch MO or a second, non-overlapping MO) should be included in Figure 6A-F. The atrium size should also be quantified.

6. The atrium, ventricle and BA size after SU5402 treatment (Figure 7C-F) should be quantified. Are the controls untreated embryos or embryos treated with DMSO vehicle only? This is not clearly indicated in the Figure Legend. The authors should also take an effort to rewrite paragraph 3, page 9 of the Results section. Showing the data in table form could facilitate the interpretation of these results.

Minor issues that should be addressed:

1. The authors should take an effort to improve the image annotations. It will also be easier for the reader to interpret certain images if key anatomical structures are delineated. E.g.:

- Supplementary Figure 1C: indicate ventricle and OFT
- Supplementary Figure 2G', 2H' and 3C'': delineate heart
- Figure 2E'', 4F', 4G', 4I', 6A', 6C', 7C'', 7D'' and 7E'' and Supplementary Figure 3E' and 3F': delineate atrium, ventricle and OFT/BA

2. In the Results section reporting the optogenetics experiments, the authors should explain the difference between "n-value" and "N-value".

3. Scale bars are missing in Figure 1A, Supplementary Figure 4A and Supplementary Figure 5D.

4. Typos:

- "red arrow" should be "red line" in Figure Legend 1B
- Figure panel labeling in Supplementary Figure Legend 3: "(D)" should be "(E)" and "(B)" should "(F)" at end of Figure Legend.
- Figure labeling on top of Supplementary Figure 4E-E'' should be "drl:mCherry, DAPI, 3 dpf" instead of "tbx1:EGFP;drl:mCherry, DAPI, 3 dpf"
- Figure Legend 5: "Red signal within the BA (asterisks)" should be "Red signal within the BA (b)"

Response to Reviewer comments: Felker et al.

Reviewer #1 (Remarks to the Author):

This is a very interesting paper that makes significant contributions to our knowledge of early heart development. In particular, it provides new insights and clarifications of previous models or hypothesis on how cells are added to the developing heart, with a focus on the ventricle and bulbus arteriosus (BA)/outflow tract. Utilizing transgenics, optigenetics and high-speed imaging, the authors identify an undifferentiated sheath of *tbx1* reporter-expressing cells that are continuously added to, and gradually differentiate at, the arterial pole. In addition, the authors find BA progenitors reside in the *tbx1* reporter-positive pharyngeal ALPM and migrate later to contribute to the late-differentiating distal pole of the ventricle to become smooth muscle. Using both FGF8 morpholinos and pharmacological inhibitors of FGF receptors, they identified a distinct temporal requirement for FGF signaling in controlling ventricle size and BA specification. Overall, this is a very strong contribution to the field, and helps pull together a wide range of previous observations into a consistent model for heart development, and helps compare heart development in zebrafish to mammals. A few points need clarification.

We thank the reviewer for her/his critical reading of our work and the provided constructive input. We have now submitted a revised version of our manuscript in which we addressed the reviewer's comments to the best of our abilities. Together with the input from the other reviewers, we appreciate how the manuscript has now improved and we hope that it is even more able to convey the key points.

Major concerns

1. Cell quantification is lacking from most of the fate-mapping imaging analysis. This should be possible for at least Figures 2 and 4.

In the analysis of co-expression of SHF marker *Isl1* and the *tbx1* transgene (Fig 2), it appears that very few cells co-express, a majority are exclusive for either *Isl1* or *tbx1* transgene. This suggests most of the SHF is not derived from *tbx1*+ cells. It would help to have a quantification from the images of how many cells (by counting) express just one or both markers.

While our work focuses on the OFT/BA in the main experiments, the reviewer raises a good point on the details of the observed *Isl1/tbx1* reporter cells. We have now performed more detailed quantification of the respective reporter expression in virtual sections of imaged embryos: these results indicate that at the IFT, the vast majority of *Isl1*+ cells are also expressing the *tbx1* reporter at time of analysis (avg. 77.3%, n=11 embryos). While the remaining avg. 22.7 of cells are only *Isl1*-positive, these cells could either be genuinely distinct from the *Isl1/tbx1* double-expressing cells, or result from the dynamic expression of the *tbx1* reporter transgene. This point warrants future detailed analysis of IFT formation using our transgene and others; we have commented on this in the revised Discussion, and have documented the quantification in revised Figure 2 together with the newly added Supplementary Video 2 depicting the cell counting.

Similarly, in contrast to marking by *drl*-transgene (Fig 2) and *myl7*-transgene (Fig 4), what percentage/how many of the ventricle and atrial cardiomyocytes, endocardial cells, etc (Fig 2 and Fig 4) express the *tbx1* transgene?

We now analyzed the percentage of *tbx1*-positive cells over the developmental series in the linear heart tube and the ventricles, as depicted in revised Figure 4G. Due to used methodologies, the quantification in the atria is not feasible, as the squamous epithelium of the atria often folds upon mounting.

Indeed, we find the *tbx1*:EGFP reporter to be weakly expressed throughout the whole endocardium; this expression domain and its connection to cranial endothelial cells is currently ongoing work in the lab for a distinct project.

2. In *fgf8a* morpholino treatment, the bulbus arteriosus (BA) and ventricle are smaller, as measured by area in the images. First, is this reduction in size due to a decreased number of cells (needs quantification) and/or size of cells? Second, is it a preferential loss of *tbx1* expressing cells or non-expressing cells? In the image (Fig 6), appears that the number of *tbx1*-transgene positive cells are the same, at least in the BA. If so, what does this have to do with the relationship of FGF function and *tbx1*-expressing lineages?

We appreciate the reviewer's constructive insight. We have now quantified the number of cells in both ventricle and BA and showed that the size difference most likely reflects the decrease in cell number, while the percentage of *tbx1*-expressing cells in the ventricle remains unaltered. The BA cells remain *tbx1*:EGFP positive, regardless of the perturbations. We now further performed control experiments to compare *fgf8a* morphants to control morpholino-injected embryos, using the generally accepted control sequence from GeneTools (i.e. guidelines proposed by Stainier et al., 2017). The corresponding quantifications for all these experiments are depicted in revised Figure 6, panels E, F,G, and J.

3. "In contrast to the mere reduction in ventricle size upon FGF inhibition, we observed a more striking and concentration-dependent effect on BA addition". This conclusion is somewhat questionable for BA development, and needs some additional analysis. It is also challenging when there is extensive edema; cells can get so stretched that they eventually die. An apoptosis assay would address that.

The reviewer raises this interesting point towards a potential alternative cause of BA loss in FGF-inhibited embryos.

We have robustly observed embryos with absent BA formation when treated with 5 μ M SU5402 from 14 ss to 22 hpf, or with 2 μ M from 14 ss until the time of observation (4 dpf) as analyzed either in *tbx1*:EGFP expressing embryos or after DAR-4M staining of the BA (Figure 7C-L, Supplementary Figure 7). Moreover, even though the analysis of embryos in Figure 5L was performed at 4 dpf, we observed BA formation in SU5402-treated embryos from 3 dpf until 5 dpf to exclude that the BA could potentially be formed and later undergo apoptosis, or a potential delay in BA formation upon FGF signaling inhibition. While control embryos showed normal BA formation, embryos with aberrant BA formation could be observed at all time points examined in the FGF signaling-inhibited cohorts.

To further examine the possibility that the extensive edema in SU5402-treated embryos through substantial stretching forces could result in apoptosis in the BA, we have now additionally included an analysis on BA formation in morphants for *tbx5a* (*heartstring*) (revised Supplementary Figure 7). In our analysis, the BA was properly formed and could be observed up to 5 dpf in *tbx5a* morphants that developed the reported severe heartstring phenotype (as well as reported milder forms of looping defects in the *tbx5a* morphants, also documented in revised Supplementary Figure 7). As the heartstring phenotype is an extreme manifestation of a stretched heart and extensive edema, yet the BA remains, we believe that the loss of the BA in FGF signaling-perturbed embryos is not likely to occur due to general apoptosis caused by the resulting edema or stretching. We have added a note on this matter also to the discussion of the manuscript.

Is it possible that the BA cells are there, and differentiated, but strong inhibition of FGF signaling prevents the expression of the *tbx1*-transgene?

We did not observe any obvious loss of *tbx1* reporter transgene activity in any of its expression domains. Moreover, to avoid any reporter bias, we assessed BA loss additionally in embryos exposed to DAR-4M to visualize the differentiated BA (Supplementary Figure 7), which led to similar results as in our analysis of the *tbx1* reporter-expressing BA.

Alternatively, this could be remedied by confocal microscopy of fixed embryo the hearts "in situ" in order to clarify.

We have now additionally included the new Supplementary Videos 8 and 9 to thoroughly visualize the loss of the BA in embryos depicted in Figure 7D,E.

Minor points

4. There are some concerns about the efficacy of CreERT2. Do you see mosaic activity that suggests inefficient activity?

Mosaicism in CreERT2 experiments is a reoccurring issue in lineage trace experiments in the field, in particular due to the relatively short timeframes of 4-OHT activity and variable sensitivity of existing *loxP* lines in zebrafish. We and others have been observing highly variable mosaicism depending on developmental time points and used CreERT2 drivers (as summarized and reviewed recently in Carney & Mosimann, 2018). Even in highly active CreERT2 drivers such as *ubi* and *drl*, we rarely observe 100% switching efficiency (i.e. Felker, Nieuwenhuize, et al., 2016; Gays et al., 2017). The images in Figure 2 and Supplementary Figures 2, 3 are representative of the overall switching efficiency we observe with *tbx1:creERT2*. Since this is indeed mosaic, we as general rule always aim to examine multiple embryos as documented in the manuscript.

5. In figure legend, mention what is insert in 1C through 1F

We have now added this information in the figure legend of Figure 1.

6. In supplemental Fig 2, for 3/3 or 4/4, it is not clear whether those different sections of the same embryo or different embryos. The level of the section (anterior-posterior axis) seems to be different, suggesting that they represent different sections of the same embryo. Also, in the heat shock control, it would help to put the exact same slice where there is a positive mark in their treated sample.

Numbers indicate individual embryos analyzed, and we now added this and more information in the figure legend. We have also added control section images for every relative position analyzed in the presented figures.

7. Merged of sup figure 4 E'' appears to be inverted (mirror image).

Now corrected.

8. Indicate number of embryos analyzed by SPIM videos.

We revisited all instances where n numbers should be mentioned, and added to the individual experiments in figure legends for videos, and to figure legends of figures that show data extracted from videos.

Reviewer #2 (Remarks to the Author):

Felker et al. use *tbx1* transgenic reporter and lineage tracing tools to image the progressive formation of heart development with a focus on the accretion of cells to form the outflow tract (bulbus arteriosus, or BA). While not fundamentally impacting the current understanding of SHF contributions to the primitive heart tube, they provide some striking images and a useful contribution to describing how *tbx1*+ cardiopharyngeal progenitors contribute to the OFT. Chiefly, they describe a “continuous progenitor sheaf” that was not fully appreciated without live imaging. Consistent with recent observations, they document nicely the contribution of pharyngeal ALPM progenitors to the OFT and demonstrate a function for FGF signaling, also consistent with previous

results. The study provides a very nice documentation of progressive OFT development and will be appreciated by the field. A few minor issues should be clarified.

We appreciate the reviewer's positive take of our paper! We have tackled the indicated issues now in our revised manuscript as outlined below, and believe that all performed revisions by the reviewers' input have greatly helped to make our work more accessible to the interested readers.

1) Several transgenic lines were generated but it was unclear which of these were used in the study. Were results validated in more than one transgenic reporter?

The reviewer raises a key point for transgenic zebrafish experiments that we also keep advocating (i.e. Mosimann & Zon, 2011; Felker & Mosimann, 2016; Carney & Mosimann, 2018), and that also reviewer 3 touched upon (see also below). We always make more than one independent transgenic line and believe this practice should be standard in the field. We had already outlined in our first submission that we had generated seven independent *tbx1:EGFP* lines, but we have now clarified this further in the text to emphasize. We added to the methods section a reference to this and in detail describe the selection procedure for the used lines. Overall, our 3.2kb *tbx1* cis-regulatory elements are among the most-robust in our collection for reproducible transgenesis with Tol2.

Were more than one CreERT line and DENDRA line also generated and did they recapitulate the normal pattern of *tbx1* expression? In particular, the CreERT line appears to label strongly the entire gut. Is this expected?

We have generated two independent lines for *tbx1:creERT2* and *tbx1:H2B-Dendra2* that provided the same results overall. We have added more details on this point in the methods section now as well. Further, we have now included images documenting the quality and faithful expression of the additional transgenic lines (revised Supplementary Figures 3,5) for completion.

Concerning the lineage labeling, it is not the entire gut that is labeled but rather the anterior parts of the endoderm-derived lineages, in particular the oral and pharynx epithelium (Supplementary Figure 2). Since early *tbx1* expression in zebrafish and mouse has been reported in anterior endoderm, our lineage tracing into anterior endoderm lineages is expected and possibly another confirmation that our reporter captures key aspects of endogenous *tbx1* expression. We have now also specified these points in the description of the relevant figures and data points.

2) Did the lineage tracing of *tbx1* pharyngeal ALPM also label head mesoderm?

The reviewer raises an interesting point that is highly relevant to the *tbx1* expression pattern. We indeed to find lineage tracing into diverse head mesoderm lineages, including cranial endothelium and different muscle groups. We have indicated these observations in Supplementary Figures 2,3 and mentioned them in the corresponding text. We hope our manuscript will also provide interesting impulses for other labs working on these lineages.

3) The temporal role of FGF signaling is unclear. BA appears to be often absent when SU was added as a pulse from 13-22 hpf. Yet the FGF8a morphants only have a reduced size of BA, even though it should be functioning through this same time frame. Is it just a hypomorph? Pulsing with SU from 24-34 hpf looks more like the morphant, with a smaller BA. How do the authors interpret this result? FGF is no longer required for the accretion of cells from the pharyngeal arches?

We thank the reviewer for raising these points for clarification. While FGF8 has been the most prevalent ligand connected to SHF development in different species, other ligands such as FGF10 have also been implicated in this process (Abu-Issa et al., 2002; Frank et al., 2002; de Pater et al., 2009; Kelly et al. 2001). Moreover, redundancies between different FGF ligands have been frequently reported in developmental processes, in particular in zebrafish (Ornitz and Itoh, 2015). Therefore, we believe that the milder defects we see in *fgf8a* morphants result from still functional components of FGF signaling in contrast to SU5402 treatments that completely abolish all FGF signaling functions.

The apparent temporal requirements for FGF signaling on BA formation noted by the reviewer are indeed highly interesting. We only observed complete failure of BA formation when FGF signaling was blocked before cardiac cone stages, but not when perturbed later during heart tube stages. As BA precursors reside in the pharyngeal mesoderm during cardiac cone stages (as shown in our opto-genetics results and previous work by several other labs), our data suggest that FGF signaling already controls smooth muscular SHF-progenitors prior to their migration towards the developing heart. We have expressed this hypothesis and its context in the discussion of Figure 7 of our manuscript.

Is reporter expression altered in PA mesoderm in the 13-22 hpf pulse?

The reviewer touches once more on an interesting point: we do not see an obvious change in the reporter levels overall, indicating no overall sensitivity of the *tbx1* reporter to perturbed FGF signaling within the timeframe of our observations. Nonetheless, the resolution of the imaging we have performed at the stages in question is likely insufficient to determine details and more subtle changes within the complex *tbx1* reporter-expressing territories.

Reviewer #3 (Remarks to the Author):

The work by Felker et al. examines the formation of the zebrafish ventricle and OFT structures by SPIM imaging with genetic and optogenetic lineage tracing, and by chemical and genetic perturbations. They show that the zebrafish ventricle forms through continuous addition from an undifferentiated *tbx1* reporter-expressing cell sheath followed by late phase accrual of the bulbus arteriosus. Furthermore, the authors report the temporal requirement for FGF signaling in controlling ventricle and BA size, and BA formation.

We appreciate the reviewer's critical input on our submitted work. In our revised manuscript version, we have now addressed all the issues to the best of our abilities, as outlined point by point below. We believe these improvements, together with the other reviewers' input, have helped to tidy up the manuscript and to document the results also better to non-zebrafish readers.

Major issues that should be addressed:

1. The selection criteria used to retain lines I, IV and V from all six transgenic lines should be mentioned. Preferentially in the Methods section.

As reviewer 2, the reviewer points out an important consideration for results presented from transgenic zebrafish (see also reply to reviewer 2 above). We invariably generate several independent lines of key transgenics to mitigate position effect-mediated artefacts that could render future transgenic work irreproducible.

We had isolated seven independent *tbx1:EGFP* lines, as already indicated in our initial submission, and we have now further emphasized and documented this point in the methods section, as suggested. We have also added more extensive and specifying text with methods. For future work by other labs that might be interested in using *tbx1:EGFP*, we strongly advocate the use of line I with official ZFIN designation *zh702*. Further, the *tbx1* cis-regulatory region we use for our transgenics seems highly inert to position effects and is among the most-consistent transgene drivers we have in our collection, facilitating the future generation of additional lines with this expression pattern.

2. The authors should show a corresponding control image for every section shown in Supplementary Figure 2A-E and 2G-K to appreciate the specificity of the GFP signal. Was the autofluorescence signal in skin and blood also present in heat shock control sections?

We fully agree with the reviewer's comment. We have now added control section images for every relative position analyzed in the presented figures. The autofluorescence was indeed also present in controls, and marked with asterisks in Supplementary Figure 2. We also observe this phenomenon in non-transgenic wildtypes, indicating that this is a general autofluorescence property of these tissues or used fixation.

3. The authors should include higher magnification images of the heart area shown in Figure 2A-A''.

We agree with the reviewer and have now added magnification panels to the revised Figure 2.

4. The Dendra2-red positive signal in Figure 5 and Supplementary Figure 6 is not always convincing as the intensity is similar to the autofluorescence signal in blood (e.g., Figure 5D and 5F and Supplementary Figure 6F and 6H). Images from unconverted control embryos captured and processed identical as the converted embryos should be shown side-by-side with the converted images at similar magnification. This will facilitate the interpretation of positive Dendra2-red signal.

We appreciate the reviewer's point on the difficulty to evaluate specific Dendra2-red signals in the images obtained from photoconverted embryos at 3.5 dpf and we apologize for this. We were similarly conflicted when assembling our material and nevertheless came to the conclusion that we will, at any cost, avoid invasive image manipulations for the sake of "cleaner" representation. Thus, our presented images represent the fluorescent signal as obtained with our imaging conditions.

Since the analysis of photoconverted cells was performed 2.5 days post-photoconversion, the remaining signal in these cells was as anticipated low, and the required high laser power and long exposure times captured autofluorescent signals in blood and other structures. Nevertheless, through the analysis of z-sections rather than just MIPs of many embryos converted in different regions of the *tbx1* reporter-expressing domain, our analyses and allocation of photoconverted cells to different parts of the heart and head at 3.5 dpf are robust. For better interpretation of our results, we have included images from unconverted control embryos captured and processed identical as the converted embryos and at the same magnification as Figure 5G,H and Supplementary Figure 6 E,J,N, as suggested by the reviewer. We also hope our data will provide a good basis for future studies in the field using *tbx1* transgenics for elucidating the contribution of the *tbx1*-expressing lineages to cranial structures.

5. A proper control group (five mismatch MO or a second, non-overlapping MO) should be included in Figure 6A-F. The atrium size should also be quantified.

We have now performed additional control experiments using the widely used control morpholino sequence by GeneTools and as widely employed in the field. The used *fgf8a* morpholino has been widely deployed in the field and is well verified for its activity in many instances, including seminal work on SHF/late-differentiating myocardium (de Pater et al., 2009). Our experiments and the controls are now represented and quantified in revised Figure 6. Due to used methodologies, the quantification of the atria is not feasible, as the squamous epithelium often folds upon mounting. We have nonetheless added more extensive quantification of the effects on overall heart and ventricle size in the revised Figure 6.

6. The atrium, ventricle and BA size after SU5402 treatment (Figure 7C-F) should be quantified. Are the controls untreated embryos or embryos treated with DMSO vehicle only? This is not clearly indicated in the Figure Legend.

We agree with reviewer and have now performed area measurements of the ventricle, atrium and BA in SU5402-treated embryos, which we have included as Figure 7F-H. We have further clarified the type of control (DMSO treated) in the figure legends of Figure 7 and Supplementary Figure 7.

The authors should also take an effort to rewrite paragraph 3, page 9 of the Results section. Showing the data in table form could facilitate the interpretation of these results.

We agree with the reviewer that the previous section was confusing. We have now completely re-worded the section and put more emphasis on the data representation in Figure 7 and Supplementary Figure 7.

Minor issues that should be addressed:

1. The authors should take an effort to improve the image annotations. It will also be easier for the reader to interpret certain images if key anatomical structures are delineated.

We agree with the reviewer and have now revised the annotations and supporting schematics for Supplementary Figure 1C; Supplementary Figure 2G', 2H', 3C''; Figure 2E'', 4F', 4G', 4I', 6A', 6C', 7C'', 7D'', 7E''; Supplementary Figure 3E', 3F'; and incorporated also the other reviewers' comments.

2. In the Results section reporting the optogenetics experiments, the authors should explain the difference between “n-value” and “N-value”.

We have now added more clear definitions of this to the results section and respective figures: n indicates number of observations, while N indicates number of performed independent experiments.

3. Scale bars are missing in Figure 1A, Supplementary Figure 4A and Supplementary Figure 5D.

We now added these scale bars in the revised figure versions.

4. Typos:

- “red arrow” should be “red line” in Figure Legend 1B
- Figure panel labeling in Supplementary Figure Legend 3: “(D)” should be “(E)” and “(B)” should “(F)” at end of Figure Legend.
- Figure labeling on top of Supplementary Figure 4E-E” should be “drl:mCherry, DAPI, 3 dpf” instead of “tbx1:EGFP;drl:mCherry, DAPI, 3 dpf”
- Figure Legend 5: “Red signal within the BA (asterisks)” should be “Red signal within the BA (b)”

These have now all been fixed and addressed in the revised manuscript, in addition to other typos we now caught upon revisiting the text.

Reviewers' Comments:

Reviewer #1:

Remarks to the Author:

This paper makes significant contributions to our knowledge of early heart development, showing how cells are added to the developing heart. In the revision, the authors have addressed all of my previous concerns, by adding of quantifications of cells or volumes to most of the fate-mapping imaging analyses, indicating numbers of experiments and embryos analyzed, adding controls, addressing mosaicism/inefficiency of CreERT2, and testing/clarifying possible alternative explanations for observations, and adding a few points to the discussion.

Reviewer #2:

Remarks to the Author:

The authors did a good job at responding to my critique and I have no further concerns. The manuscript is improved and experiments are rigorous, with conclusions justified.

Reviewer #3:

Remarks to the Author:

The authors have adequately addressed all of the concerns that I raised.

Response to Reviewer comments: Felker et al., revision

Reviewer #1 (Remarks to the Author):

This paper makes significant contributions to our knowledge of early heart development, showing how cells are added to the developing heart. In the revision, the authors have addressed all of my previous concerns, by adding of quantifications of cells or volumes to most of the fate-mapping imaging analyses, indicating numbers of experiments and embryos analyzed, adding controls, addressing mosaicism/inefficiency of CreERT2, and testing/clarifying possible alternative explanations for observations, and adding a few points to the discussion.

We are delighted that our revisions have addressed all of the reviewer's concerns, and we thank the reviewer once more for the critical input that has greatly improved the manuscript.

Reviewer #2 (Remarks to the Author):

The authors did a good job at responding to my critique and I have no further concerns. The manuscript is improved and experiments are rigorous, with conclusions justified.

We thank the reviewer for her/his positive take on our revisions, and appreciate the input on our initial submission.

Reviewer #3 (Remarks to the Author):

The authors have adequately addressed all of the concerns that I raised.

We are happy that our revisions have addressed all the reviewer's concerns, and thank the reviewer again for the constructive input.